# An Investigation into Whitening Loss for Self-supervised Learning

**Xi Weng**[*1]**, Lei Huang**[*✉ 1,2]**, Lei Zhao**[*1]**,**
**Rao Muhammad Anwer**[2]**, Salman Khan**[2]**, Fahad Shahbaz Khan**[2]
[1]SKLSDE, Institute of Artificial Intelligence, Beihang University, Beijing, China
[2]Mohamed bin Zayed University of Artificial Intelligence, UAE

## Abstract

A desirable objective in self-supervised learning (SSL) is to avoid feature collapse. Whitening loss guarantees collapse avoidance by minimizing the distance between embeddings of positive pairs under the conditioning that the embeddings from different views are whitened. In this paper, we propose a framework with an informative indicator to analyze whitening loss, which provides a clue to demystify several interesting phenomena as well as a pivoting point connecting to other SSL methods. We reveal that batch whitening (BW) based methods do not impose whitening constraints on the embedding, but they only require the embedding to be full-rank. This full-rank constraint is also sufficient to avoid dimensional collapse. Based on our analysis, we propose channel whitening with random group partition (CW-RGP), which exploits the advantages of BW-based methods in preventing collapse and avoids their disadvantages requiring large batch size. Experimental results on ImageNet classification and COCO object detection reveal that the proposed CW-RGP possesses a promising potential for learning good representations. The code is available at https://github.com/winci-ai/CW-RGP.

## 1 Introduction

Self-supervised learning (SSL) has made significant progress over the last several years [1, 19, 6, 16, 8], almost reaching the performance of supervised baselines on many downstream tasks [33, 24, 35]. Several recent approaches rely on a joint embedding architecture in which a dual pair of networks are trained to produce similar embeddings for different views of the same image [8]. Such methods aim to learn representations that are invariant to transformation of the same input. One main challenge with the joint embedding architectures is how to prevent a *collapse* of representation, in which the two branches ignore the inputs and produce identical and constant output representations [6, 8].

One line of work uses contrastive learning methods that attract different views from the same image (positive pairs) while pull apart different images (negative pairs), which can prevent constant outputs from the solution space [43]. While the concept is simple, these methods need large batch size to obtain a good performance [19, 6, 37]. Another line of work tries to directly match the positive targets without introducing negative pairs. A seminal approach, BYOL [16], shows that an extra predictor and momentum is essential for representation learning. SimSiam [8] further generalizes [16] by empirically showing that stop-gradient is essential for preventing trivial solutions. Recent works generalize the collapse problem into *dimensional collapse* [21, 25][2] where the embedding vectors only span a lower-dimensional subspace and would be highly correlated. Therefore, the embedding vector dimensions would vary together and contain redundant information. To prevent the *dimensional*

---

[*]equal contribution    [✉]corresponding author (huangleiAI@buaa.edu.cn). This work was partially done while Lei Huang was a visiting scholar at Mohamed bin Zayed University of Artificial Intelligence, UAE.

[2]This collapse is also referred to *informational collapse* in [2].

*collapse*, whitening loss is proposed by only minimizing the distance between embeddings of positive pairs under the condition that embeddings from different views are whitened [12, 21]. A typical way is using batch whitening (BW) and imposing the loss on the whitened output [12, 21], which obtains promising results.

Although whitening loss has theoretical guarantee in avoiding collapse, we experimentally observe that this guarantee depends on which kind of whitening transformation [26] is used in practice (see Section 3.2 for details). This interesting observation challenges the motivations of whitening loss for SSL. Besides, the motivation of whitening loss is that the whitening operation can remove the correlation among axes [21] and a whitened representation ensures the examples scattered in a spherical distribution [12]. Based on this argument, one can use the whitened output as the representation for downstream tasks, but it is not used in practice. To this end, this paper investigates whitening loss and tries to demystify these interesting observations. Our contributions are as follows:

- We decompose the symmetric formulation of whitening loss into two asymmetric losses, where each asymmetric loss requires an online network to match a whitened target. This mechanism provides a pivoting point connecting to other methods, and a way to understand why certain whitening transformation fails to avoid *dimensional collapse*.

- Our analysis shows that BW based methods do not impose whitening constraints on the embedding, but they only require the embedding to be full-rank. This full-rank constraint is also sufficient to avoid *dimensional collapse*.

- We propose channel whitening with random group partition (CW-RGP), which exploits the advantages of BW-based method in preventing collapse and avoids their disadvantages requiring large batch size. Experimental results on ImageNet classification and COCO object detection show that CW-RGP has promising potential in learning good representation.

## 2 Related Work

A desirable objective in self-supervised learning is to avoid feature collapse.

**Contrastive learning** prevents collapse by attracting positive samples closer, and spreading negative samples apart [43, 44]. In these methods, negative samples play an important role and need to be well designed [34, 1, 20]. One typical mechanism is building a memory bank with a momentum encoder to provide consistent negative samples, proposed in MoCos [19], yielding promising results [19, 7, 9, 30]. Other works include SimCLR [6] addresses that more negative samples in a batch with strong data augmentations perform better. Contrastive methods require large batch sizes or memory banks, which tends to be costly, promoting the questions whether negative pairs is necessary.

**Non-contrastive methods** aim to accomplish SSL without introducing negative pairs explicitly [3, 4, 31, 16, 8]. One typical way to avoid representational collapse is the introduction of asymmetric network architecture. BYOL [16] appends a predictor after the online network and introduce momentum into the target network. SimSiam [8] further simplifies BYOL by removing the momentum mechanism, and shows that stop-gradient to target network serves as an alternative approximation to the momentum encoder. Other progress includes an asymmetric pipeline with a self-distillation loss for Vision Transformers [5]. It remains not clear how the asymmetric network avoids collapse without negative pairs, leaving the debates on batch normalization (BN) [14, 41, 36] and stop-gradient [8, 46], even though preliminary works have attempted to analyze the training dynamics theoretical with certain assumptions [40] and build a connection between asymmetric network with contrastive learning methods [39]. Our work provides a pivoting point connecting asymmetric network to profound whitening loss in avoiding collapse.

**Whitening loss** has theoretical guarantee in avoiding collapse by minimizing the distance of positive pairs under the conditioning that the embeddings from different views are whitened [45, 12, 21, 2]. One way to obtain whitened output is imposing a whitening penalty as regularization on embedding—the so-called soft whitening, which is proposed in Barlow Twins [45], VICReg [2] and CCA-SSG [47]. Another way is using batch whitening (BW) [22]—the so-called hard whitening, which is used in W-MSE [12] and Shuffled-DBN [21]. We propose a different hard whitening method—channel whitening (CW) that has the same function that ensures all the singular values of transformed output being one for avoiding collapse. But CW is more numerical stable and works better when batch size is small, compared to BW. Furthermore, our CW with random group partition (CW-RGP) can effectively control the extent of constraint on embedding and obtain better performance in practice. We note that a recent work ICL [48] proposes to decorrelate instances, like CW but having several significant

differences in technical details. ICL uses "stop-gradient" for the whitening matrix, while CW requires back-propagation through the whitening transformation. Besides, ICL uses extra pre-conditioning on the covariance and whitening matrices, which is essential for the numerical stability, while CW does not use extra pre-conditioning and can work well since it encourages the embedding to be full-rank.

## 3 Exploring Whitening Loss for SSL

### 3.1 Preliminaries

Let $\mathbf{x}$ denote the input sampled uniformly from a set of images $\mathbb{D}$, and $\mathbb{T}$ denote the set of data transformations available for augmentation. We consider the Siamese network $f_\theta(\cdot)$ parameterized by $\theta$. It takes as input two randomly augmented views, $\mathbf{x}_1 = \mathcal{T}_1(\mathbf{x})$ and $\mathbf{x}_2 = \mathcal{T}_2(\mathbf{x})$, where $\mathcal{T}_{1,2} \in \mathbb{T}$. The network $f_\theta(\cdot)$ is trained with an objective function that minimizes the distance between embeddings obtained from different views of the same image:

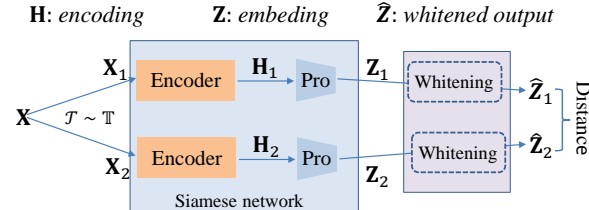

Figure 1: The basic notations for SSL used in this paper.

$$\mathcal{L}(\mathbf{x}, \theta) = \mathbb{E}_{\mathbf{x} \sim \mathbb{D}, \, \mathcal{T}_{1,2} \sim \mathbb{T}} \; \ell\big(f_\theta(\mathcal{T}_1(\mathbf{x})), f_\theta(\mathcal{T}_2(\mathbf{x}))\big). \tag{1}$$

where $\ell(\cdot, \cdot)$ is a loss function. In particular, the Siamese network usually consists of an encoder $E_{\theta_e}(\cdot)$ and a projector $G_{\theta_g}(\cdot)$. Their outputs $\mathbf{h} = E_{\theta_e}(\mathcal{T}(\mathbf{x}))$ and $\mathbf{z} = G_{\theta_g}(\mathbf{h})$ are referred to as *encoding* and *embedding*, respectively. We summarize the notations and use the corresponding capital letters denoting mini-batch data in Figure 1. Under this notation, we have $f_\theta(\cdot) = G_{\theta_g}(E_{\theta_e}(\cdot))$ with learnable parameters $\theta = \{\theta_e, \theta_g\}$. The *encoding* $\mathbf{h}$ is usually used as representation for evaluation by either training a linear classifier [19] or transferring to downstream tasks. This is due to that $\mathbf{h}$ is shown to obtain significantly better performance than the *embedding* $\mathbf{z}$ [6, 8].

The mean square error (MSE) of $L_2-$normalized vectors is usually used as the loss function [8]:

$$\ell(\mathbf{z}_1, \mathbf{z}_2) = \|\frac{\mathbf{z}_1}{\|\mathbf{z}_1\|_2} - \frac{\mathbf{z}_2}{\|\mathbf{z}_2\|_2}\|_2^2, \tag{2}$$

where $\| \cdot \|_2$ denotes the $L_2$ norm. This loss is also equivalent to the negative cosine similarity, up to a scale of $\frac{1}{2}$ and an optimization irrelevant constant.

**Collapse and Whitening Loss.** While minimizing Eqn. 1, a trivial solution known as *collapse* could occur such that $f_\theta(\mathbf{x}) \equiv \mathbf{c}, \; \forall \mathbf{x} \in \mathbb{D}$. The state of *collapse* will provide no gradients for learning and offer no information for discrimination. Moreover, a weaker collapse condition called *dimensional collapse* can be easily arrived, for which the projected features collapse into a low-dimensional manifold. As illustrated in [21], dimensional collapse is associated with strong correlations between axes, which motivates the use of whitening method in avoiding the dimensional collapse. The general idea of whitening loss [12] is to minimize Eqn. 1, under the condition that *embeddings* from different views are whitened, which can be formulated as[3]:

$$\min_\theta \mathcal{L}(\mathbf{x}; \theta) = \mathbb{E}_{\mathbf{x} \sim \mathbb{D}, \, \mathcal{T}_{1,2} \sim \mathbb{T}} \; \ell(\mathbf{z}_1, \mathbf{z}_2),$$

$$s.t. \; cov(\mathbf{z}_i, \mathbf{z}_i) = \mathbf{I}, \; i \in \{1, 2\}. \tag{3}$$

Whitening loss provides theoretical guarantee in avoiding (dimensional) collapse, since the embedding is whitened with all axes decorrelated [12, 21]. While it is difficult to directly solve the problem of Eqn. 3, Ermolov *et al.* [12] propose to whiten the mini-batch embedding $\mathbf{Z} \in \mathbb{R}^{d_z \times m}$ using batch whitening (BW) [22, 38] and impose the loss on the whitened output $\widehat{\mathbf{Z}} \in \mathbb{R}^{d_z \times m}$, given the mini-batch inputs $\mathbf{X}$ with size of $m$, as follows:

$$\min_\theta \mathcal{L}(\mathbf{X}; \theta) = \mathbb{E}_{\mathbf{X} \sim \mathbb{D}, \, \mathcal{T}_{1,2} \sim \mathbb{T}} \; \|\widehat{\mathbf{Z}}_1 - \widehat{\mathbf{Z}}_2\|_F^2$$

$$with \; \widehat{\mathbf{Z}}_i = \Phi(\mathbf{Z}_i), \; i \in \{1, 2\}, \tag{4}$$

where $\Phi(\cdot)$ denotes the whitening transformation over mini-batch data.

---

[3]The dual view formulation can be extended to $s$ different views, as shown in [12].

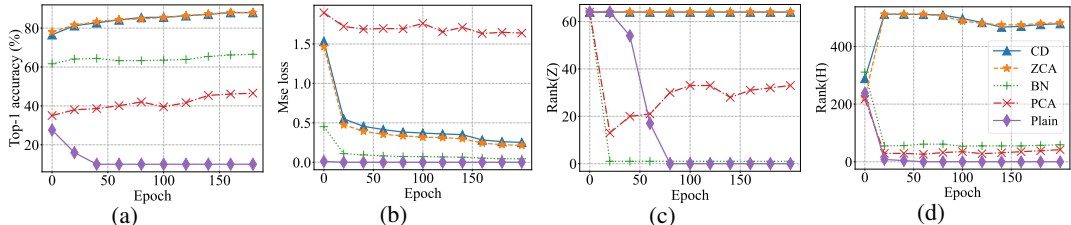

Figure 2: Effects of different whitening transformations for SSL. We use the ResNet-18 as the encoder (dimension of representation is 512.), a two layer MLP with ReLU and BN appended as the projector (dimension of embedding is 64). The model is trained on CIFAR-10 for 200 epochs with batch size of 256 and standard data argumentation, using Adam optimizer [27] (more details of experimental setup please see *supplementary materials*). We show (a) the linear evaluation accuracy; (b) the training loss; (c) the rank of *embedding*; (d) the rank of *encoding*.

**Whitening Transformations.**    There are an infinite number of possible whitening matrices, as shown in [26, 22], since any whitened data with a rotation is still whitened. For simplifying notation, we assume $\mathbf{Z}$ is centered by $\mathbf{Z} := \mathbf{Z}(\mathbf{I} - \frac{1}{m}\mathbf{1}\mathbf{1}^T)$. Ermolov *et al.* [12] propose W-MSE that uses Cholesky decomposition (CD) whitening: $\Phi_{CD}(\mathbf{Z}) = \mathbf{L}^{-1}\mathbf{Z}$ in Eqn. 4, where $\mathbf{L}$ is a lower triangular matrix from the Cholesky decomposition, with $\mathbf{L}\mathbf{L}^T = \Sigma$. Here $\Sigma = \frac{1}{m}\mathbf{Z}\mathbf{Z}^T$ is the covariance matrix of the *embedding*. Hua *et al.* [21] use zero-phase component analysis (ZCA) whitening [22] in Eqn. 4: $\Phi_{ZCA} = \mathbf{U}\Lambda^{-\frac{1}{2}}\mathbf{U}^T$, where $\Lambda = \mathrm{diag}(\lambda_1, \ldots, \lambda_{d_z})$ and $\mathbf{U} = [\mathbf{u}_1, ..., \mathbf{u}_{d_z}]$ are the eigenvalues and associated eigenvectors of $\Sigma$, *i.e.*, $\mathbf{U}\Lambda\mathbf{U}^T = \Sigma$. Another famous whitening is principal components analysis (PCA) whitening: $\Phi_{PCA} = \Lambda^{-\frac{1}{2}}\mathbf{U}^T$ [26, 22].

## 3.2  Empirical Investigation on Whitening Loss

In this section, we conduct experiments to investigate the effects of different whitening transformations $\Phi(\cdot)$ used in Eqn. 4 for SSL. Besides, we investigate the performances of different features (including *encoding* $\mathbf{H}$, *embedding* $\mathbf{Z}$ and the whitened output $\widehat{\mathbf{Z}}$) used as representation for evaluation. For illustration, we first define the *rank* and *stable-rank* [42] of a matrix as follows:

**Definition 1.**  *Given a matrix $\mathbf{A} \in \mathbb{R}^{d \times m}, d \leq m$, we denote $\{\lambda_1, ..., \lambda_d\}$ the singular values of $\mathbf{A}$ in a descent order with convention $\lambda_1 > 0$. The **rank** of $\mathbf{A}$ is the number of its non-zero singular values, denoted as $Rank(\mathbf{A}) = \sum_{i=1}^d \mathbb{I}(\lambda_i > 0)$, where $\mathbb{I}(\cdot)$ is the indicator function. The **stable-rank** of $\mathbf{A}$ is denoted as $r(\mathbf{A}) = \frac{\sum_{i=1}^d \lambda_i}{\lambda_1}$.*

By definition, $Rank(\mathbf{A})$ can be a good indicator to evaluate the extent of *dimensional collapse* of $\mathbf{A}$, and $r(\mathbf{A})$ can be an indicator to evaluate the extent of whitening of $\mathbf{A}$. It can be demonstrated that $r(\mathbf{A}) \leq Rank(\mathbf{A}) \leq d$ [42]. Note that if $\mathbf{A}$ is fully whitened with covariance matrix $\mathbf{A}\mathbf{A}^T = m\mathbf{I}$, we have $r(\mathbf{A}) = Rank(\mathbf{A}) = d$. We also define normalized rank as $\widehat{Rank}(\mathbf{A}) = \frac{Rank(\mathbf{A})}{d}$ and normalized stable-rank as $\widehat{r}(\mathbf{A}) = \frac{r(\mathbf{A})}{d}$, for comparing the extent of *dimensional collapse* and whitening of matrices with different dimensions, respectively.

**PCA Whitening Fails to Avoid Dimensional Collapse.**    We compare the effects of ZCA, CD, PCA transformations for whitening loss, evaluated on CIFAR-10 using the standard setup for SSL (see Section 4.1 for details). Besides, we also provide the result of batch normalization (BN) that only performs standardization without decorrelating the axes, and the 'Plain' method that imposes the loss directly on *embedding*. From Figure 2, we observe that naively training a Siamese network ('Plain') results in collapse both on the *embedding* (Figure 2(c)) and *encoding* (Figure 2(d)), which significantly hampers the performance (Figure 2(a)), although its training loss becomes close to zero (Figure 2(b)). We also observe that an extra BN imposed on the *embedding* prevents collapse to a point. However, it suffers from the dimensional collapse where the rank of *embedding* and *encoding* are significantly low, which also hampers the performance. ZCA and CD whitening both maintain high rank of *embedding* and *encoding* by decorrelating the axes, ensuring high linear evaluation accuracy. However, we note that PCA whitening shows significantly different behaviors: PCA whitening cannot decrease the loss and even cannot avoid the dimensional collapse, which also leads to significantly downgraded performance. This interesting observation challenges the motivations of whitening loss for SSL. We defer the analyses and illustration in Section 3.3.

**Whitened Output is not a Good Representation.**    As introduced before, the motivation of whitening loss for SSL is that the whitening operation can remove the correlation among axes [21] and a

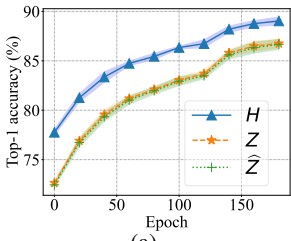 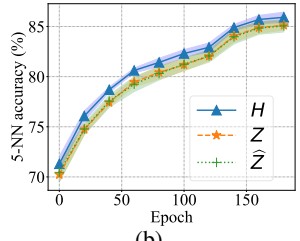 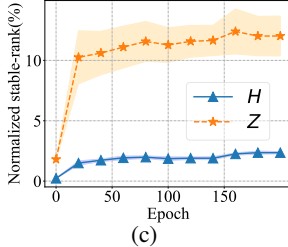

|(a)|(b)|(c)|

Figure 3: Comparisons of features when using *encoding* $\mathbf{H}$, *embedding* $\mathbf{Z}$ and whitened output $\widehat{\mathbf{Z}}$ respectively. We follow the same experimental setup as Figure 2. We show (a) the linear evaluation accuracy; (b) the kNN accuracy; (c) the normalized stable-rank for comparing the extent of whitening (note that the normalized stable-rank of $\widehat{\mathbf{Z}}$ is always $100\%$ during training and we omit it for clarity). The results are averaged by five random seeds, with standard deviation shown using shaded region.

whitened representation ensures that the examples scattered in a spherical distribution [12], which is sufficient to avoid collapse. Based on this argument, one should use the whitened output $\widehat{\mathbf{Z}}$ as the representation for downstream tasks, rather than the *encoding* $\mathbf{H}$ that is commonly used. This raises questions that whether $\mathbf{H}$ is well whitened and whether the whitened output is a good feature. We conduct experiments to compare the performances of whitening loss, when using $\mathbf{H}$, $\mathbf{Z}$ and $\widehat{\mathbf{Z}}$ as representations for evaluation respectively. The results are shown in Figure 3. We observe that using whitened output $\widehat{\mathbf{Z}}$ as a representation has significantly worse performance than using $\mathbf{H}$. Furthermore, we find that the normalized stable rank of $\mathbf{H}$ is significantly smaller than $100\%$, which suggests that $\mathbf{H}$ is not well whitened. These results show that the whitened output could not be a good representation.

### 3.3 Analysing Decomposition of Whitening Loss

For clarity, we use the mini-batch input with size of $m$. Given one mini-batch input $\mathbf{X}$ with two augmented views, Eqn. 4 can be formulated as:

$$\mathcal{L}(\mathbf{X}) = \frac{1}{m}\|\widehat{\mathbf{Z}}_1 - \widehat{\mathbf{Z}}_2\|_F^2. \tag{5}$$

Let us consider a proxy loss described as:

$$\mathcal{L}^{'}(\mathbf{X}) = \underbrace{\frac{1}{m}\|\widehat{\mathbf{Z}}_1 - (\widehat{\mathbf{Z}}_2)_{st}\|_F^2}_{\mathcal{L}_1^{'}} + \underbrace{\frac{1}{m}\|(\widehat{\mathbf{Z}}_1)_{st} - \widehat{\mathbf{Z}}_2\|_F^2}_{\mathcal{L}_2^{'}}, \tag{6}$$

where $(\cdot)_{st}$ indicates the stop-gradient operation. It is easy to demonstrate that $\frac{\partial \mathcal{L}}{\partial \theta} = \frac{\partial \mathcal{L}^{'}}{\partial \theta}$ (see *supplementary materials* for proof). That is, the optimization dynamics of $\mathcal{L}$ is equivalent to $\mathcal{L}^{'}$. By looking into the first term of Eqn. 6, we have:

$$\mathcal{L}_1^{'} = \frac{1}{m}\|\phi(\mathbf{Z}_1)\mathbf{Z}_1 - (\widehat{\mathbf{Z}}_2)_{st}\|_F^2. \tag{7}$$

Here, we can view $\phi(\mathbf{Z}_1)$ as a predictor that depends on $\mathbf{Z}_1$ during forward propagation, and $\widehat{\mathbf{Z}}_2$ as a whitened target with $r(\widehat{\mathbf{Z}}_2) = Rank(\widehat{\mathbf{Z}}_2) = d_z$. In this way, we find that minimizing $\mathcal{L}_1^{'}$ only requires the embedding $\mathbf{Z}_1$ being full-rank with $Rank(\widehat{\mathbf{Z}}_1) = d_z$, as stated by following proposition.

**Proposition 1.** *Let $\mathbb{A} = \arg min_{\mathbf{Z}_1} \mathcal{L}_1^{'}(\mathbf{Z}_1)$. We have that $\mathbb{A}$ is not an empty set, and $\forall \mathbf{Z}_1 \in \mathbb{A}$, $\mathbf{Z}_1$ is full-rank. Furthermore, for any $\{\sigma_i\}_{i=1}^{d_z}$ with $\sigma_1 \geq \sigma_2 \geq, ..., \sigma_{d_z} > 0$, we construct $\widetilde{\mathbb{A}} = \{\mathbf{Z}_1|\mathbf{Z}_1 = \mathbf{U}_2 \, diag(\sigma_1, \sigma_2, ..., \sigma_{d_z}) \, \mathbf{V}_2^T$, where $\mathbf{U}_2 \in \mathbb{R}^{d_z \times d_z}$ and $\mathbf{V}_2 \in \mathbb{R}^{m \times d_z}$ are from the singular value decomposition of $\widehat{\mathbf{Z}}_2$, i.e., $\mathbf{U}_2(\sqrt{m}\mathbf{I})\mathbf{V}_2^T = \widehat{\mathbf{Z}}_2$. When we use ZCA whitening, we have $\widetilde{\mathbb{A}} \subseteq \mathbb{A}$.*

The proof is shown in *supplementary materials*. Proposition 1 states that there are infinity matrix with full-rank that is the optimum when minimizing $\mathcal{L}_1^{'}$ w.r.t. $\mathbf{Z}_1$. Therefore, minimizing $\mathcal{L}_1^{'}$ only requires the embedding $\mathbf{Z}_1$ being full-rank with $Rank(\widehat{\mathbf{Z}}_1) = d_z$, and does not necessarily impose the constraints on $\mathbf{Z}_1$ to be whitened with $r(\mathbf{Z}_1) = d_z$. Similar analysis also applies to $\mathcal{L}_2^{'}$ and

minimizing $\mathcal{L}_2^{'}$ requires $\mathbf{Z}_2$ being full-rank. Therefore, BW-based methods shown in Eqn. 4 do not impose whitening constraints on the embedding as formulated in Eqn. 3, but they only require the embedding to be **full-rank**. This full-rank constraint is also sufficient to avoid dimensional collapse for embedding, even though it is a weaker constraint than whitening.

Our analysis further implies that whitening loss in its symmetric formulation (Eqn. 5) can be decomposed into two asymmetric losses (Eqn. 6), where each asymmetric loss requires an online network to match a whitened target. This mechanism provides a pivot connecting to other methods, and a clue to understand why PCA whitening fails to avoid dimensional collapse for SSL.

**Connection to Asymmetric Methods.** The asymmetric formulation of whitening loss shown in Eqn. 7 bears resemblance to those asymmetry methods without negative pairs, *e.g.*, SimSiam [8]. In these methods, an extra predictor is incorporated and the stop-gradient is essential for avoid collapse. In particular, SimSiam uses the objective as:

$$\mathcal{L}(\mathbf{X}) = \frac{1}{m}\|P_{\theta_p}(\cdot) \circ \mathbf{Z}_1 - (\mathbf{Z}_2)_{st}\|_F^2 + \frac{1}{m}\|P_{\theta_p}(\cdot) \circ \mathbf{Z}_2 - (\mathbf{Z}_1)_{st}\|_F^2, \tag{8}$$

where $P_{\theta_p}(\cdot)$ is the predictor with learnable parameters $\theta_p$. By contrasting Eqn. 7 and the first term of Eqn. 8, we find that: 1) BW-based whitening loss ensures a whitened target $\widehat{\mathbf{Z}}_2$, while SimSiam does not put constraint on the target $\mathbf{Z}_2$; 2) SimSiam uses a learnable predictor $P_{\theta_p}(\cdot)$, which is shown to empirically avoid collapse by matching the rank of the covariance matrix by back-propagation [40], while BW-based whitening loss has an implicit predictor $\phi(\mathbf{Z}_1)$ depending on the input itself, which is a full-rank matrix by design. Based on this analysis, we find that BW-based whitening loss can surely avoid collapse if the loss converges well, while Simsian can not provide such a guarantee in avoiding collapse. Similar analysis also applies to BYOL [16], except that BYOL uses a momentum target network for providing target signal.

**Connection to Soft Whitening.** VICReg [2] also encourages whitened *embedding* produced from different views, but by imposing a whitening penalty as a regularization on the *embedding*, which is called soft whitening. In particular, given a mini-batch input, the objective of VICReg is as follows[4]:

$$\mathcal{L}(\mathbf{X}) = \frac{1}{m}\|\mathbf{Z}_1 - \mathbf{Z}_2\|_F^2 + \alpha \sum_{i=1}^{2}(\|\frac{1}{m}\mathbf{Z}_i\mathbf{Z}_i^T - \lambda\mathbf{I}\|_F^2), \tag{9}$$

where $\alpha \geq 0$ is the penalty factor. Similarly, we can use a proxy loss for VICReg and considering its term corresponding to optimizing $\mathbf{Z}_1$ only (similar to Eqn. 7), we have:

$$\mathcal{L}_{VICReg}^{'}(\mathbf{X}) = \frac{1}{m}\|\mathbf{Z}_1 - (\mathbf{Z}_2)_{st}\|_F^2 + \alpha\|\frac{1}{m}\mathbf{Z}_1\mathbf{Z}_1^T - \lambda\mathbf{I}\|_F^2. \tag{10}$$

Based on this formulation, we observe that VICReg requires *embedding* $\mathbf{Z}_1$ to be whitened by, 1) the additional whitening penalty, and 2) fitting the (expected) whitened targets $\mathbf{Z}_2$. By contrasting Eqns. 7 and 10, we highlight that the so-called hard whitening methods, like W-MSE [12], only impose full-rank constraints on the embedding, while soft whitening methods indeed impose whitening constraints. Similar analysis also applies to Barlow Twins [45], except that the whitening/decorrelation penalty is imposed on the cross-covariance matrix of embedding from different views.

**Connection to Other Non-contrastive Methods.** SwAV [4], a clustering-based method, uses a "swapped" prediction mechanism where the cluster assignment (code) of a view is predicted from the representation of another view, by minimizing the following objective:

$$\mathcal{L}(\mathbf{X}) = \ell(\mathbf{C}^T\mathbf{Z}_1, (\mathbf{Q}_2)_{st}) + \ell(\mathbf{C}^T\mathbf{Z}_2, (\mathbf{Q}_1)_{st}). \tag{11}$$

Here, $\mathbf{C}$ is the prototype matrix learned by back-propagation, $\mathbf{Q}_i$ is the predicted code with equal-partition and high-entropy constraints, and SwAV uses cross-entropy loss as $\ell(\cdot, \cdot)$ to match the distributions. The constraints on $\mathbf{Q}_i$ are approximately satisfied during optimization, by using the iterative Sinkhorn-Knopp algorithm conditioned on the input $\mathbf{C}^T\mathbf{Z}_i$. Note that SwAV explicitly uses stop-gradient when it calculates the target $\mathbf{Q}_i$. By contrasting Eqn. 7 and the first term of Eqn. 11, we find that: 1) SwAV can be viewed as an online network to match a target with constraints, like BW-based whitening loss, even thought the constraints imposed on the targets between them are

---

[4]Note the slight difference where VICReg uses margin loss on the diagonal of covariance, while our notation uses MSE loss.

different; 2) From the perspective of asymmetric structure, SwAV indeed uses a linear predictor $\mathbf{C}^T$ that is also learned by back-propagation like SimSiam, while BW-based whitening loss has an implicit predictor $\phi(\mathbf{Z}_1)$ depending on the input itself. Similar analysis also applies to DINO [5], which further simplifies the formulation of SwAV by removing the prototype matrix and directly matching the output of another view, from the view of knowledge distillation. DINO uses centering and sharpening operations to impose the constraints on the target (output of another view). One significant difference between DINO and whitening loss is that DINO uses population statistics of centering calculated by moving average, while whitening loss uses the mini-batch statistics of whitening.

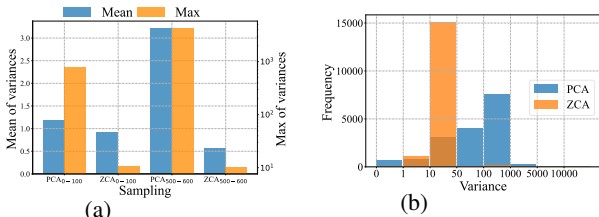

(a)  (b)

**Why PCA Whitening Fails to Avoid Dimensional Collapse?** Based on Eqn. 7, we note that whitening loss can favorably provide full-rank constraints on the embedding under the condition that the online network can match the whitened targets well. We experimentally find that PCA-based whitening loss provides volatile sequence of whitened targets during training, as shown in Figure 4(a). It is difficult for the online network to match such a target signal with significant variation, resulting in minimal decrease in the whitening loss (see Figure 2). Furthermore, we observe that PCA-based whitening loss has also significantly varying whitening matrix sequences $\{\phi^t(\cdot)\}$ (Figure 4(b)), even given the same input data. This coincides with the observation

Figure 4: Illustration of PCA-based whitening loss suffering from training instability. We follow the same experimental setup as Figure 2. Given a certain mini-batch input ($m = 2048$), we monitor its whitened output $\widehat{\mathbf{Z}}^t$ and whitening matrix $\Phi^t$ for each epoch $t$. We calculate the variance along the training epochs for each element of $\widehat{\mathbf{Z}}$ and $\Phi$. We show (a) the mean and maximum of variances of $\widehat{\mathbf{Z}}$, noting that $PCA_{0-100}$ indicates the variance of PCA whitened output is calculated along the first 100 epochs; and (b) the histogram of variance of $\Phi$.

in [16, 8], where an unstable predictor results in significant degenerate performance. Our observations are also in accordance with the arguments in [22, 23] that PCA-based BW shows significantly large stochasticity. We note that ZCA whitening can provide relatively stable sequences of whitened targets and whitening matrix during training (Figure 4), which ensures stable training for SSL. This is likely due to the property of ZCA-based whitening that minimizes the total squared distance between the original and whitened variables [26, 22].

**Why Whitened Output is not a Good Representation?** A whitened output removes the correlation among axes [21] and ensures the examples scattered in a spherical distribution [12], which bears resemblance to contrastive learning where different examples are pulled away. We conduct experiments to compare SimCLR [6], BYOL [16], VICReg [2] and W-MSE [12], and monitor the cosine similarity for all negative pairs, stable-rank and rank during training. From Figure 5, we find that all methods can achieve a high rank on the *encoding*. This is driven by the improved extent of whitening on the *embedding*. Furthermore, we observe that the negatives cosine similarity decreases during the training, while the extent of stable-rank increases, for all methods. This observation suggests that a representation with stronger extent of whitening is more likely to have less similarity among different examples. We further conduct experiments to validate this argument, using VICReg

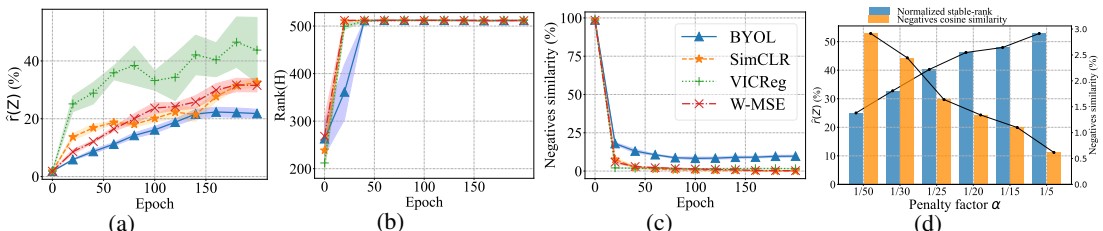

(a)  (b)  (c)  (d)

Figure 5: Comparison of different SSL methods. We follow the same experimental setup as Figure 2. We show (a) the normalized stable-rank of *embedding*; (b) the rank of *encoding*; (c) the negatives cosine similarity, calculated on the *embeddings* from all negative pairs (different examples). We also train VICReg with varying penalty factor $\alpha$ to show the relationship between the normalized stable-rank and negatives cosine similarity in (d). Here, we use embedding dimension of 64. We have similar observations when using the embedding dimension of other numbers (*e.g.*, 128 and 256).

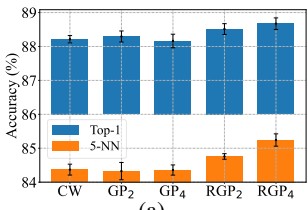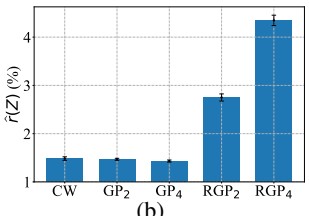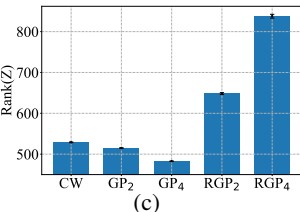

(a)             (b)             (c)

Figure 6: Illustration of CW with random group partition. We follow the same experimental setup as Figure 2, except that we set the dimension of embedding as 2048 tailored for CW. We use 'GP$_2$' ('RGP$_2$') to indicate CW using group partition (random group partition), with a group number of 2. (a) The linear and k-NN accuracies; (b) The normalized stable-rank of embedding; (c) The rank of embedding. All experiments are repeated five times, with standard deviation shown as error bars.

with varying penalty factor $\alpha$ (Eqn. 10) to adjust the extent of whitening on *embedding* (Figure 5(d)). Therefore, a whitened output leads to the state that all examples have dissimilar features. This state can break the potential manifold the examples in the same class belong to, which makes the learning more difficult [17]. Similar analysis for contrastive learning is also shown in [6], where classes represented by the projected output (*embedding*) are not well separated, compared to *encoding*.

## 4 Channel Whitening with Random Group Partition

One main weakness of BW-based whitening loss is that the whitening operation requires the number of examples (mini-batch size) $m$ to be larger than the size of channels $d$, to avoid numerical instability[5]. This requirement limits its usage in scenarios where large batch of training data cannot be fit into the memory. Based on previous analysis, the whitening loss can be viewed as an online learner to match a whitened target with all singular values being one. We note the key of whitening loss is that it conducts a transformation $\phi : \mathbf{Z} \to \widehat{\mathbf{Z}}$, ensuring that the singular values of $\widehat{\mathbf{Z}}$ are one. We thus propose channel whitening (CW) that ensures the examples in a mini-batch are orthogonal:

$$Centering : \mathbf{Z}_c = (\mathbf{I} - \frac{1}{d}\mathbf{1} \cdot \mathbf{1}^T)\mathbf{Z}, \qquad Whitening : \widehat{\mathbf{Z}} = \mathbf{Z}_c\Phi, \qquad (12)$$

where $\Phi \in \mathbb{R}^{m \times m}$ is the 'whitening matrix' that is derived from the corresponding 'covariance matrix': $\Sigma' = \frac{1}{d-1}\mathbf{Z}_c^T\mathbf{Z}_c$. In our implementation, we use ZCA whitening to obtain $\Phi$. CW ensures the examples in a mini-batch are orthogonal to each other, with $\widehat{\mathbf{Z}}^T\widehat{\mathbf{Z}} = \frac{1}{d-1}\mathbf{I}$. This means CW has the same ability as BW for SSL in avoiding the dimensional collapse, by providing target $\widehat{\mathbf{Z}}$ whose singular values are one. More importantly, one significant advantage of CW is that it can obtain numerical stability when the batch size is small, since the condition that $d > m$ can be obtained by design (*e.g.*, we can set the channel number of embedding $d$ to be larger than the batch size $m$). Besides, we find that CW can amplify the full-rank constraints on the embedding by dividing the channels/neurons into random groups, as we will illustrate.

**Random Group Partition.** Given the embedding $\mathbf{Z} \in \mathbb{R}^{d \times m}, d > m$, we divide it into $g \geq 1$ groups $\{\mathbf{Z}^{(i)} \in \mathbb{R}^{\frac{d}{g} \times m}\}_{i=1}^{g}$, where we assume that $d$ is divisible by $g$ and ensure $\frac{d}{g} > m$. We then perform CW on each $\mathbf{Z}^{(i)}, i = 1, ..., g$. Note that the ranks of $\mathbf{Z}$ and $\mathbf{Z}^{(i)}$ are all at most $m$. Therefore, CW with group partition provides $g$ constraints with $Rank(\mathbf{Z}^{(i)}) = m$ on embedding, compared to CW without group partition that only one constraint with $Rank(\mathbf{Z}) = m$. Although CW with group partition can provide more full-rank constraints for mini-batch data, we find that it can also make the population data correlated, if group partition is all the same during training, which decreases the rank and does not improve the performance in accuracy by our experiments (Figure 6). We find random group partition, which randomly divide the channels/neurons into group for each iteration (mini-batch data), can alleviate this issue and obtain an improved performance, from Figure 6. We call our method as channel whitening with random group partition (CW-RGP), and provide the full algorithm and PyTorch-style code in *supplementary materials*.

We note that Hua *et al.* [21] use a similar idea for BW, called Shuffled-DBN. However Shuffled-DBN cannot well amplify the full-rank constraints by using more groups, since BW-based methods require $m > \frac{d}{g}$ to avoid numerical instability. We further show that CW-RGP works remarkably better than

---

[5]An empirical setting is $m = 2d$ that can obtain good performance as shown in [12, 21].

Table 1: Classification accuracy (top 1) of a linear classifier and a 5-nearest neighbors classifier for different loss functions and datasets with a ResNet-18 encoder.

| Method | CIFAR-10 | | CIFAR-100 | | STL-10 | | Tiny-ImageNet | |
|---|---|---|---|---|---|---|---|---|
| | linear | 5-nn | linear | 5-nn | linear | 5-nn | linear | 5-nn |
| SimCLR [6] | 91.80 | 88.42 | 66.83 | 56.56 | 90.51 | 85.68 | 48.84 | 32.86 |
| BYOL [16] | 91.73 | 89.45 | 66.60 | 56.82 | 91.99 | 88.64 | **51.00** | **36.24** |
| SimSiam [8] (repro.) | 90.51 | 86.82 | 66.04 | 55.79 | 88.91 | 84.84 | 48.29 | 34.21 |
| Shuffled-DBN [21] (repro.) | 90.45 | 88.15 | 66.07 | 56.97 | 89.20 | 84.51 | 48.60 | 32.14 |
| Barlow Twins [45] (repro.) | 88.51 | 86.53 | 65.78 | 55.76 | 88.36 | 83.71 | 47.44 | 32.65 |
| VICReg [2] (repro.) | 90.32 | 88.41 | 66.45 | 56.78 | 90.78 | 85.72 | 48.71 | 33.35 |
| Zero-ICL [48] (repro.) | 88.12 | 86.64 | 61.91 | 53.47 | 86.35 | 82.51 | 46.25 | 32.74 |
| W-MSE 2 [12] | 91.55 | 89.69 | 66.10 | 56.69 | 90.36 | 87.10 | 48.20 | 34.16 |
| W-MSE 4 [12] | 91.99 | 89.87 | 67.64 | 56.45 | 91.75 | 88.59 | 49.22 | 35.44 |
| CW-RGP 2 (ours) | 91.92 | 89.54 | 67.51 | 57.35 | 90.76 | 87.34 | 49.23 | 34.04 |
| CW-RGP 4 (ours) | **92.47** | **90.74** | **68.26** | **58.67** | **92.04** | **88.95** | 50.24 | 35.99 |

Shuffled-DBN in the subsequent experiments. We attribute this results to the ability of CW-RGP in amplifying the full-rank constraints by using groups.

## 4.1 Experiments for Empirical Study

In this section, we conduct experiments to validate the effectiveness of our proposed CW-RGP. We evaluate the performances of CW-RGP for classification on CIFAR-10, CIFAR-100 [28], STL-10 [10], TinyImageNet [29] and ImageNet [11]. We also evaluate the effectiveness in transfer learning, for a pre-trained model using CW-RGP. We run the experiments on one workstation with 4 GPUs. For more details of implementation and training protocol, please refer to *supplementary materials*.

**Evaluation for Classification**  We first conduct experiments on small and medium size datasets (including CIFAR-10, CIFAR-100, STL-10 and Tiny-ImageNet), strictly following the setup of *W-MSE* paper [12]. Our CW-RGP inherits the advantages

Table 2: Comparisons on ImageNet linear classification. All are based on ResNet-50 encoder. The table is mostly inherited from [8].

| Method | Batch size | 100 eps | 200 eps |
|---|---|---|---|
| SimCLR [6] | 4096 | 66.5 | 68.3 |
| MoCo v2 [7] | 256 | 67.4 | 69.9 |
| BYOL [16] | 4096 | 66.5 | 70.6 |
| SwAV [4] | 4096 | 66.5 | 69.1 |
| SimSiam [8] | 256 | 68.1 | 70.0 |
| W-MSE 4 [12] | 4096 | 69.4 | - |
| Zero-CL [48] | 1024 | 68.9 | - |
| BYOL [16] (repro.) | 512 | 66.1 | 69.2 |
| SwAV [4] (repro.) | 512 | 65.8 | 67.9 |
| W-MSE 4 [12] (repro.) | 512 | 66.7 | 67.9 |
| **CW-RGP 4 (ours)** | 512 | **69.7** | **71.0** |

of W-MSE in exploiting different views. CW-RGP 2 and CW-RGP 4 indicate our methods with $s = 2$ and $s = 4$ positive views extracted per image respectively, similar to *W-MSE* [12]. The results of baselines shown in Table. 1 are partly inherited in [12], except that we reproduce certain baselines under the same training and evaluation settings as in [12] (some different hyper-parameter settings are shown in *supplementary materials*). We observe that CW-RGP obtains the highest accuracy on almost all the datasets except Tiny-ImageNet. Besides, CW-RGP with 4 views are generally better than 2, similar to W-MSE. These results show that CW-RGP is a competitive SSL method. We also confirm that CW with random group partition could obtain a higher performance than BW (and with random group partition), comparing CW-RGP to W-MSE and Shuffled-DBN.

We then conduct experiments on large-scale ImageNet, strictly following the setup of SimSiam paper [8]. The results of baselines shown in Table 2 are mostly reported in [8], except that the result of W-MSE 4 is from the W-MSE paper [12] and we reproduce BYOL [16], SwAV [4] and W-MSE 4 [12] under a batch size of 512 based on the same training and evaluation settings as in [8] for fairness. CW-RGP 4 is trained with a batch size of 512 and gets the highest accuracy among all methods under both 100 and 200 epochs training. We find that our CW-RGP can also work well when combined with the whitening penalty used in VICReg. Note that we also try a batch size of 256 under 100-epoch training, which gets the top-1 accuracy of $69.5\%$.

**Transfer to downstream tasks**  We examine the representation quality by transferring our model to other tasks, including VOC [13] object detection, COCO [32] object detection and instance segmentation. We use the baseline (except for the pre-training model, the others are exactly the same) of the detection codebase from MoCo [19] for CW-RGP to produce the results. The results of baselines shown in Table3 are mostly inherited from [8]. We clearly observe that CW-RGP performs better than or on par with these state-of-the-art approaches on COCO object detection and instance segmentation, which shows the great potential of CW-RGP in transferring to downstream tasks.

Table 3: Transfer Learning. All competitive unsupervised methods are based on 200-epoch pre-training in ImageNet (IN). The table is mostly inherited from [8]. Our CW-RGP is performed with 3 random seeds, with mean and standard deviation reported.

| Method | VOC 07+12 detection | | | COCO detection | | | COCO instance seg. | | |
|---|---|---|---|---|---|---|---|---|---|
| | $AP_{50}$ | AP | $AP_{75}$ | $AP_{50}$ | AP | $AP_{75}$ | $AP_{50}$ | AP | $AP_{75}$ |
| Scratch | 60.2 | 33.8 | 33.1 | 44.0 | 26.4 | 27.8 | 46.9 | 29.3 | 30.8 |
| IN-supervised | 81.3 | 53.5 | 58.8 | 58.2 | 38.2 | 41.2 | 54.7 | 33.3 | 35.2 |
| SimCLR [6] | 81.8 | 55.5 | 61.4 | 57.7 | 37.9 | 40.9 | 54.6 | 33.3 | 35.3 |
| MoCo v2 [7] | **82.3** | 57.0 | 63.3 | 58.8 | 39.2 | 42.5 | 55.5 | 34.3 | 36.6 |
| BYOL [16] | 81.4 | 55.3 | 61.1 | 57.8 | 37.9 | 40.9 | 54.3 | 33.2 | 35.0 |
| SwAV [4] | 81.5 | 55.4 | 61.4 | 57.6 | 37.6 | 40.3 | 54.2 | 33.1 | 35.1 |
| SimSiam [8] | 82.0 | 56.4 | 62.8 | 57.5 | 37.9 | 40.9 | 54.2 | 33.2 | 35.2 |
| **CW-RGP (ours)** | $82.2_{\pm0.07}$ | $\mathbf{57.2}_{\pm0.10}$ | $\mathbf{63.8}_{\pm0.11}$ | $\mathbf{60.5}_{\pm0.28}$ | $\mathbf{40.7}_{\pm0.14}$ | $\mathbf{44.1}_{\pm0.14}$ | $\mathbf{57.3}_{\pm0.16}$ | $\mathbf{35.5}_{\pm0.12}$ | $\mathbf{37.9}_{\pm0.14}$ |

**Ablation for Random Group Partition.** We also conduct experiments to show the advantages of random group partition for channel whitening. We use 'CW', 'CW-GP' and 'CW-RGP' to indicate channel whitening without group partition, with group partition and with random group partition, respectively. We further consider the setup with $s = 2$ and $s = 4$ positive views. We use the same setup as in Table 1 and show the results in Table 4. We have similar observation as in Figure 6 that CW with random group partition improves the performance.

Table 4: Results of ablation for random group partition.

| Method | CIFAR-10 | | CIFAR-100 | |
|---|---|---|---|---|
| | linear | 5-nn | linear | 5-nn |
| CW 2 | 91.66 | 88.99 | 66.26 | 56.36 |
| CW-GP 2 | 91.61 | 88.89 | 66.17 | 56.53 |
| **CW-RGP 2** | **91.92** | **89.54** | **67.51** | **57.35** |
| CW 4 | 92.10 | 90.12 | 66.90 | 57.12 |
| CW-GP 4 | 92.08 | 90.06 | 67.34 | 57.28 |
| **CW-RGP 4** | **92.47** | **90.74** | **68.26** | **58.67** |

**Ablation for Batch Size.** Here, we conduct experiments to empirically show the advantages of CW over BW, in terms of the stability using different batch size. We train CW and BW on ImageNet-100, using batch size ranging in $\{32, 64, 128, 256\}$. Figure 7 shows the results. We can find that CW is more robust for small batch size training.

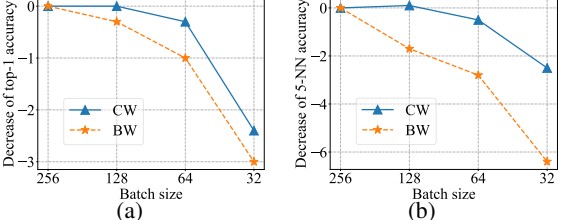

Figure 7: Decrease in top-1 and 5-nn accuracy (in % points) of CW and BW at 100 epochs on ImageNet-100.

## 5 Conclusion and Limitation

In this paper, we invested whitening loss for SSL, and observed several interesting phenomena with further clarification based on our analysis framework. We showed that batch whitening (BW) based methods only require the embedding to be full-rank, which is also a sufficient condition for collapse avoidance. We proposed channel whitening with random group partition (CW-RGP) that is well motivated theoretically in avoiding a collapse and has been validated empirically in learning good representation.

**Limitation.** Our work only shows how to avoid collapse by using whitening loss, but does not explicitly show what should be the extent of whitening of a good representation. We note that a concurrent work addresses this problem by connecting the eigenspectrum of a representation to a power law [15], and shows the coefficient of the power law is a strong indicator for the effects of representation. We believe our work can be further extended when combined with the analyses from [15]. Besides, our work does not answer how the projector affects the extents of whitening between *encoding* and *embedding* [18], which is important to answer why *encoding* is usually used as a representation for evaluation, rather than the whitened output or *embedding*. Our attempts, shown in *supplementary materials*, provide preliminary results, but does not offer an answer to this question.

**Acknowledgement** This work was partially supported by the National Key Research and Development Plan of China under Grant 2021ZD0112901, National Natural Science Foundation of China (Grant No. 62106012), the Fundamental Research Funds for the Central Universities.

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
