# Supplementary Materials: An Investigation into Whitening Loss for Self-supervised Learning

**Xi Weng**[*1], **Lei Huang**[*✉ 1,2], **Lei Zhao**[*1],
**Rao Muhammad Anwer**[2], **Salman Khan**[2], **Fahad Shahbaz Khan**[2]
[1]SKLSDE, Institute of Artificial Intelligence, Beihang University, Beijing, China
[2]Mohamed bin Zayed University of Artificial Intelligence, UAE

## A  Details of Experimental Setup for Small and Medium Size Datasets

In this section, we provide the details of the implementation and training protocol for the experiments on small and medium size datasets (including CIFAR-10, CIFAR-100, STL-10, Tiny-ImageNet and ImageNet-100). Our implementation is based on the released codebase of *W-MSE* [7][2].

### A.1  Datasets

The followings are the descriptions of 5 small and medium scale datasets, commonly used to evaluate the effectiveness of SSL models.

• CIFAR-10 and CIFAR-100 [13], two small-scale datasets composed of $32 \times 32$ images with 10 and 100 classes, respectively.

• STL-10 [4], derived from ImageNet [6], with $96 \times 96$ resolution images and more than 100K training samples.

• Tiny ImageNet [14], a reduced version of ImageNet [6], composed of 200 classes with images scaled down to $64 \times 64$. The total number of images is: 100K (training) and 10K (testing).

• ImageNet-100 [16], a random 100-class subset of ImageNet [6].

### A.2  Analytical Experiments

In section 3 of the submitted paper, we conduct several experiments on CIFAR-10 to illustrate our analysis. We provide a brief description of the setup in the caption of Figure 2 of the submitted paper. Here, we describe the details of these experiments. All experiments are uniformly based on the following training settings, unless otherwise stated in the figures of the submitted paper.

**Training Settings**  We use the ResNet-18 as the encoder (dimension of *encoding* is 512.), a two layer MLP with ReLU and BN appended as the projector (dimension of the hidden layer and embedding are 1024 and 64 respectively). The model is trained on CIFAR-10 for 200 epochs with a batch size of 256, using Adam optimizer [12] with a learning rate of $3 \times 10^{-3}$, and learning rate warm-up for the first 500 iterations and a 0.2 learning rate drop at the last 50 and 25 epochs. The weight decay is set as $10^{-6}$. All transformations are performed with 2 positives extracted per image with standard data argumentation (see Section A.3 for details). We use the same evaluation protocol as in *W-MSE* [7].

**Method Settings**  In the experiments shown in Figure 2 and Figure 4 of the paper, fully PCA whitening suffers the dimensional collapse and further produces numerical instability. Therefore, we

---

[*]equal contribution  [✉]corresponding author (huangleiAI@buaa.edu.cn). This work was partially done while Lei Huang was a visiting scholar at Mohamed bin Zayed University of Artificial Intelligence, UAE.

[2]https://github.com/htdt/self-supervised

use the MSE loss without L2 normalization and partition its channel dimension into 4 groups, which makes it possible to finish training in a normal way. Despite the reduced setting for whitening, PCA whitening still has problems as shown in Figure 2 and Figure 4. In other experiments, we use the MSE loss with L2 normalization for all methods, as *W-MSE* [7] does.

### A.3   Experimental Setup for Comparison of Baselines

In section 4.1 of the paper, we compare our channel whitening with random group partition (CW-RGP) to the state-of-the-art SSL methods on CIFAR-10, CIFAR-100, STL-10 and Tiny-ImageNet datasets. Here, we describe the training details of these experiments. Except for the hyper-parameters relating to CW-RGP itself (*e.g.*, the group number), our experimental setups are strictly following the setup of the W-MSE [7] paper, as following descriptions.

**Encoder and Projector**   We use the ResNet-18 [11] as the encoder and the dimension of *encoding* is 512. We use a 2-layers MLP as the projector: one hidden layer with BN and Relu applied to it and a linear layer as output. In the experiments of CIFAR-10, CIFAR-100 and STL-10, the dimension of the hidden layer in the projector and *embedding* are 1024 and 512, respectively. In the experiments of Tiny-ImageNet, the dimension of the hidden layer in the projector and *embedding* are 2048 and 1024, respectively.

**Image Transformation Details**   We make the image transformation following the details in [2], which extract crops with a random size from 0.2 to 1.0 of the original area and a random aspect ratio from 3/4 to 4/3 of the original aspect ratio. The horizontal mirroring is applied with a probability of 0.5. The color jittering configuration is (0.4, 0.4, 0.4, 0.1) with a probability of 0.8 and grayscaling with a probability of 0.1. For ImageNet-100, the crop size is from 0.08 to 1.0, jittering is strengthened to (0.8, 0.8, 0.8, 0.2), grayscaling probability is 0.2, and Gaussian blurring is with a probability of 0.5. We use only one crop at testing time in all the experiments (standard protocol).

**Optimizer and Learning Rate Schedule**   We use the Adam optimizer [12]. We apply the same number of epochs and learning rate schedule to all the compared methods. Specifically, for CIFAR-10 and CIFAR-100, we use 1,000 epochs with a learning rate of $3 \times 10^{-3}$; for STL-10, 2,000 epochs with a learning rate of $2 \times 10^{-3}$; for Tiny-ImageNet, 1000 epochs with a learning rate of $2 \times 10^{-3}$. In these experiments, we use a 0.2 learning rate drop at the last 50 and 25 epochs. The weight decay is $10^{-6}$. In all experiments, we use learning rate warm-up for the first 500 iterations of the optimizer. We use a batch size of 512 for CW-RGP 2 in CIFAR-100, STL-10 and Tiny ImageNet experiments, while 256 for the others.

**Evaluation Protocol**   We use the same setup of evaluation protocol as in *W-MSE* [7]: training the linear classifier for 500 epochs using the Adam optimizer and the labeled training set of each specific dataset, without data augmentation; the learning rate is exponentially decayed from $10^{-2}$ to $10^{-6}$ and the weight decay is $5 \times 10^{-6}$. In addition, we also evaluate the accuracy of a k-nearest neighbors classifier (k-NN, k = 5) in these experiments.

For our CW-RGP, we denote 'RGP$_2$' to indicate CW using random group partition, with a group number of 2. We find that our CW-RGP can also work well when batch-slicing, proposed in W-MSE [7], is used. We thus also use batch slicing (a default setting in the released code of W-MSE) to ensure that the channel number is larger than the batch size for our CW-RGP. In the experiments of CIFAR-10, CIFAR-100 and STL-10, we use RGP$_4$ and the slicing sub-batch size is 64. In the experiments of Tiny-ImageNet, we use RGP$_2$ and the slicing sub-batch size is 128.

For a fair comparison, we also reproduce several related methods (including SimSiam [3], Barlow Twins [17], VICReg [1], and Zero-ICL [18]) under the same training and evaluation settings as in [7]. However, These methods using the configuration of hyper-parameters based on the original baselines in [7] get poor results in our training mode (e.g, a 2-layers projector, using Adam optimizer [12], 'step' learning rate schedule, and so on). In the reproduction experiments, we use the recommended 2048-2048-2048 projector for these methods which obtains significantly better results than the default 1024-512 or 1024-64 projector in [7]. In particular, for Barlow Twins [17], we set the trade-off coefficient $\lambda$ to 0.0078 instead of the recommended 0.0051 in [17], since that using the recommended 0.0051 has a significant degenerated performance in our experiments. For SimSiam [3], we use the SGD optimizer and cosine learning rate schedule as recommended in [3], because the loss value

fluctuates sharply and it leads to very poor results when we use Adam optimizer for the training. We conjecture that the optimization mechanism of Adam may be not suitable for the training of predictor in SimSiam [3]. Other settings not mentioned here are the same as in [7] by default.

## B  Proofs

### B.1  proof of $\frac{\partial \mathcal{L}}{\partial \theta} = \frac{\partial \mathcal{L}'}{\partial \theta}$.

As stated in Section 3.3 of the paper, given one mini-batch input $\mathbf{X}$ with two augmented views, we say the loss:

$$\mathcal{L}(\mathbf{X}) = \frac{1}{m}\|\widehat{\mathbf{Z}}_1 - \widehat{\mathbf{Z}}_2\|_F^2. \tag{1}$$

and the proxy loss:

$$\mathcal{L}'(\mathbf{X}) = \frac{1}{m}\|\widehat{\mathbf{Z}}_1 - (\widehat{\mathbf{Z}}_2)_{st}\|_F^2 + \frac{1}{m}\|(\widehat{\mathbf{Z}}_1)_{st} - \widehat{\mathbf{Z}}_2\|_F^2, \tag{2}$$

has the same gradients *w.r.t.* the learnable parameters $\theta$, *i.e.*, $\frac{\partial \mathcal{L}}{\partial \theta} = \frac{\partial \mathcal{L}'}{\partial \theta}$. Here, we provide the proof.

*Proof.* Note that $\widehat{\mathbf{Z}}_1$ and $\widehat{\mathbf{Z}}_2$ are the function of $\theta$. Based on the chain rule, we have:

$$
\begin{aligned}
\frac{\partial \mathcal{L}}{\partial \theta} &= \frac{\partial \frac{1}{m}\|\widehat{\mathbf{Z}}_1 - \widehat{\mathbf{Z}}_2\|_F^2}{\partial \theta} \\
&= \frac{1}{m}\frac{\partial \|\widehat{\mathbf{Z}}_1 - \widehat{\mathbf{Z}}_2\|_F^2}{\partial \widehat{\mathbf{Z}}_1}\frac{\partial \widehat{\mathbf{Z}}_1}{\partial \theta} + \frac{1}{m}\frac{\partial \|\widehat{\mathbf{Z}}_1 - \widehat{\mathbf{Z}}_2\|_F^2}{\partial \widehat{\mathbf{Z}}_2}\frac{\partial \widehat{\mathbf{Z}}_2}{\partial \theta}.
\end{aligned}
\tag{3}
$$

Similarly, we have:

$$
\begin{aligned}
\frac{\partial \mathcal{L}'}{\partial \theta} &= \frac{\partial (\frac{1}{m}\|\widehat{\mathbf{Z}}_1 - (\widehat{\mathbf{Z}}_2)_{st}\|_F^2 + \frac{1}{m}\|(\widehat{\mathbf{Z}}_1)_{st} - \widehat{\mathbf{Z}}_2\|_F^2)}{\partial \theta} \\
&= \frac{1}{m}\frac{\partial \|\widehat{\mathbf{Z}}_1 - (\widehat{\mathbf{Z}}_2)_{st}\|_F^2}{\partial \widehat{\mathbf{Z}}_1}\frac{\partial \widehat{\mathbf{Z}}_1}{\partial \theta} + \frac{1}{m}\frac{\partial \|(\widehat{\mathbf{Z}}_1)_{st} - \widehat{\mathbf{Z}}_2\|_F^2}{\partial \widehat{\mathbf{Z}}_2}\frac{\partial \widehat{\mathbf{Z}}_2}{\partial \theta} \\
&= \frac{1}{m}\frac{\partial \|\widehat{\mathbf{Z}}_1 - \widehat{\mathbf{Z}}_2\|_F^2}{\partial \widehat{\mathbf{Z}}_1}\frac{\partial \widehat{\mathbf{Z}}_1}{\partial \theta} + \frac{1}{m}\frac{\partial \|\widehat{\mathbf{Z}}_1 - \widehat{\mathbf{Z}}_2\|_F^2}{\partial \widehat{\mathbf{Z}}_2}\frac{\partial \widehat{\mathbf{Z}}_2}{\partial \theta}.
\end{aligned}
\tag{4}
$$

From Eqn. 3 and Eqn. 4, we have $\frac{\partial \mathcal{L}}{\partial \theta} = \frac{\partial \mathcal{L}'}{\partial \theta}$. □

### B.2  Proof of Proposition 1

As stated in Section 3.3 of the paper, by looking into the first term of Eqn. 2, we aim to minimize the following objective:

$$\min_{\mathbf{Z}_1}\mathcal{L}'_1(\mathbf{Z}_1) = \min_{\mathbf{Z}_1}\frac{1}{m}\|\phi(\mathbf{Z}_1)\mathbf{Z}_1 - (\widehat{\mathbf{Z}}_2)_{st}\|_F^2. \tag{5}$$

Here, $\phi(\mathbf{Z}_1)$ is the whitening matrix that depends on $\mathbf{Z}_1$, and $\widehat{\mathbf{Z}}_2$ is a whitened matrix with $\frac{1}{m}\widehat{\mathbf{Z}}_2\widehat{\mathbf{Z}}_2^T = \mathbf{I}$. We prove the following Proposition.

**Proposition 1.** *Let $\mathbb{A} = \arg min_{\mathbf{Z}_1}\mathcal{L}'_1(\mathbf{Z}_1)$. We have that $\mathbb{A}$ is not an empty set, and $\forall \mathbf{Z}_1 \in \mathbb{A}$, $\mathbf{Z}_1$ is full-rank. Furthermore, for any $\{\sigma_i\}_{i=1}^{d_z}$ with $\sigma_1 \geq \sigma_2 \geq, ..., \sigma_{d_z} > 0$, we construct $\widetilde{\mathbb{A}} = \{\mathbf{Z}_1|\mathbf{Z}_1 = \mathbf{U}_2 \, diag(\sigma_1, \sigma_2, ..., \sigma_{d_z}) \, \mathbf{V}_2^T$, where $\mathbf{U}_2 \in \mathbb{R}^{d_z \times d_z}$ and $\mathbf{V}_2 \in \mathbb{R}^{m \times d_z}$ are from the singular value decomposition of $\widehat{\mathbf{Z}}_2$, i.e., $\mathbf{U}_2(\sqrt{m}\mathbf{I})\mathbf{V}_2^T = \widehat{\mathbf{Z}}_2$. When we use ZCA whitening, we have $\widetilde{\mathbb{A}} \subseteq \mathbb{A}$.*

*Proof.* Based on the fact that $\mathcal{L}'_1 \geq 0$, we have $\mathbb{A} = \{\mathbf{Z}_1|\mathcal{L}'_1(\mathbf{Z}_1) = 0\}$. It is easy to validate that $\mathcal{L}'_1(\widehat{\mathbf{Z}}_2) = 0$, and we have $\widehat{\mathbf{Z}}_2 \in \mathbb{A}$. Therefore, $\mathbb{A}$ is not an empty set.

We then prove that $\forall \mathbf{Z}_1 \in \mathbb{A}$, $\mathbf{Z}_1$ is full-rank. We assume that for any $\mathbf{Z}_1 \in \mathbb{A}$ and $\mathbf{Z}_1$ is not a full-rank matrix, *i.e.*, $Rank(\mathbf{Z}_1 < d_z)$. We have $Rank(\phi(\mathbf{Z}_1)\mathbf{Z}_1) \leq Rank(\mathbf{Z}_1) < d_z$. We thus

have that $\phi(\mathbf{Z}_1)\mathbf{Z}_1$ is not a full-rank matrix. Therefore, it is impossible for $\phi(\mathbf{Z}_1)\mathbf{Z}_1 = \widehat{\mathbf{Z}}_2$ since $\widehat{\mathbf{Z}}_2$ is a full-rank matrix. So $\mathcal{L}'_1(\mathbf{Z}_1) > 0$, which is contradictory to $\mathbf{Z}_1 \in \mathbb{A}$. Therefore, we have $\forall \mathbf{Z}_1 \in \mathbb{A}$, $\mathbf{Z}_1$ is full-rank

For any $\{\sigma_i\}_{i=1}^{d_z}$ with $\sigma_1 \geq \sigma_2 \geq, ..., \sigma_{d_z} > 0$, let $\mathbf{Z}_1 = \mathbf{U}_2 \operatorname{diag}(\sigma_1, \sigma_2, ..., \sigma_{d_z}) \mathbf{V}_2^T$, we now prove that $\phi(\mathbf{Z}_1)\mathbf{Z}_1 = \widehat{\mathbf{Z}}_2$ when using ZCA whitening. We know $\phi(\mathbf{Z}_1) = \Phi_{ZCA} = \mathbf{U}\Lambda^{-\frac{1}{2}}\mathbf{U}^T$, where $\Lambda = \operatorname{diag}(\lambda_1, \ldots, \lambda_{d_z})$ and $\mathbf{U} = [\mathbf{u}_1, ..., \mathbf{u}_{d_z}]$ are the eigenvalues and associated eigenvectors of the covariance matrix $\Sigma$ of $\mathbf{Z}_1$. We know $\Sigma = \frac{1}{m}\mathbf{Z}_1\mathbf{Z}_1^T = \mathbf{U}_2 \operatorname{diag}(\sigma_1^2/m, \sigma_2^2/m, ..., \sigma_{d_z}^2/m) \mathbf{U}_2^T$. Since the eigen decomposition of $\Sigma$ is unique, we have $\phi(\mathbf{Z}_1) = \mathbf{U}_2 \operatorname{diag}(\sqrt{m}/\sigma_1, \sqrt{m}/\sigma_2, ..., \sqrt{m}/\sigma_{d_z})$ $\mathbf{U}_2^T$. Therefore, $\phi(\mathbf{Z}_1)\mathbf{Z}_1 = \mathbf{U}_2 \operatorname{diag}(\sqrt{m}/\sigma_1, \sqrt{m}/\sigma_2, ..., \sqrt{m}/\sigma_{d_z}) \mathbf{U}_2^T\mathbf{U}_2 \operatorname{diag}(\sigma_1, \sigma_2, ..., \sigma_{d_z})$ $\mathbf{V}_2^T = \mathbf{U}_2(\sqrt{m}\mathbf{I})\mathbf{V}_2^T = \widehat{\mathbf{Z}}_2$. We thus have $\widetilde{\mathbb{A}} \subseteq \mathbb{A}$.

$\square$

## C    Algorithm of CW-RPG

We describe our CW-RGP algorithm in Py-Torch style code, shown in Figure I.

```python
def channel_whitening(x, g, eps=0):
    # x: input feature with size [m, d] or [m, d, H, W]
    # g: the group number of group whitening
    x_flatten = x.view(x.size()[0], -1)
    f_dim = x_flatten.size()[-1]
    shuffle = torch.randperm(f_dim).tolist()
    # centering
    mean = x_flatten.mean(-1, keepdim=True)
    x_centered = x_flatten - mean
    # random group partition
    x_group = x_centered[:, shuffle].reshape(x.size()[0], g, -1).permute(1, 2, 0)
    f_cov = torch.bmm(x_group.permute(0, 2, 1), x_group) / (x_group.shape[1] - 1)
    eye = torch.eye(x.size(0)).type(x.type()).reshape(1, x.size(0), x.size(0)).repeat(g, 1, 1)
    # compute whitening matrix
    sigma = (1 - eps) * f_cov + eps * eye
    u, eig, _ = sigma.svd()
    scale = eig.rsqrt()
    wm = torch.bmm(u, torch.diag_embed(scale))
    wm = torch.bmm(wm, u.permute(0, 2, 1))
    # whiten
    decorrelated = torch.bmm(x_group, wm)
    shuffle_recover = [shuffle.index(i) for i in range(f_dim)]
    decorrelated = decorrelated.permute(2, 0, 1).reshape(-1, f_dim)[:, shuffle_recover]

    output = decorrelated.view_as(x)
    return output
```

Figure I: Py-Torch style code of CW-RPG.

# D Details of Experiments for Large-Scale Classification and Transfer Learning

In this section, we provide the details of implementation and training protocol for the experiments on large-scale ImageNet [6] classification, and transfer learning to VOC [8] object detection, COCO [15] object detection and instance segmentation.

## D.1 Datasets

• ImageNet [6], the well-known largescale dataset with about 1.3M training images and 50K test images, spanning over 1000 classes.

• VOC07+12 [8], the PASCAL Visual Object Classes Challenge. VOC2007: 20 classes with 9,963 images containing 24,640 annotated objects; VOC2012: 20 classes with 11,530 images containing 27,450 ROI annotated objects and 6,929 segmentations.

• COCO2017 [15], a large-scale object detection, segmentation, and captioning dataset with 330K images containing 1.5 million object instances.

## D.2 Experiment on ImageNet

In section 4.1 of the paper, we compare our CW-RGP to the state-of-the-art SSL methods on large-scale ImageNet classification. Here, we describe the training details of these experiments. Our implementation is based on the released codebase of *SimSiam* [3][3]. Except for the hyper-parameters relating to CW-RGP itself, we strictly follow the setup of the SimSiam paper [3].

**Encoder and Projector**    We use the ResNet-50 [11] as the encoder and the dimension of *encoding* is 2048. We use a 3-layers MLP as the projector: two hidden layers with BN and Relu applied to it and a linear layer as output. The dimension of the hidden layer and embedding are 2048 and 1024, respectively.

**Image Transformation Details**    In image transformation, we follow the details in [3]: crop size from 0.2 to 1.0, no strengthened jittering (0.4, 0.4, 0.4, 0.1) with probability 0.8, grayscaling probability 0.2, and Gaussian blurring with 0.5 probability. We use standard protocol at testing time.

**Optimizer and Learning Rate Schedule**    We apply the SGD optimizer, using a learning rate of *lr* × BatchSize / 256 with a base *lr* of 0.05 and cosine decay schedule. The weight decay is $10^{-4}$ and the SGD momentum is 0.9. In addition, we use learning rate warm-up for the first 500 iterations of the optimizer. We only try the batch size of 256 and 512 due to memory limitation.

**Evaluation Protocol**    We use the same setup of evaluation protocol as in *Simsiam* [3]: training the *linear classifier* for 100 epochs with the *LARS* optimizer (using a learning rate of *lr* × BatchSize / 256 with a base *lr* of 0.1 and cosine decay schedule). The batch size for evaluation is 1024.

For our CW-RGP, we use $RGP_2$ for CW. We find that our CW-RGP can also work well when combined with the whitening penalty (covariance loss) used in VICReg [1]. For adapting to the sample orthogonalization in CW, we use a covariance loss along the channel dimension (see Section F for details). We empirically set the weight of covariance loss as 0.001 for the half training epochs to amplify the extent of whitening, which obtains a top-1 accuracy of 69.7% when training 100 epochs, compared to 69.6% of the method using CW-RGP only. Here, we address that our CW-RGP can combine with covariance loss to obtain good results. We believe the performance can be further improved, if we fine-tune the weight of covariance loss.

## D.3 Experiments for Transfer Learning

In this part, we describe the training details of experiments for transfer learning. Our implementation is based on the released codebase of *MoCo* [10][4] for transfer learning to object detection and instance

---

[3]*https://github.com/facebookresearch/simsiam* under the CC-BY-NC 4.0 license.
[4]*https://github.com/facebookresearch/moco/tree/main/detection* under the CC-BY-NC 4.0 license.

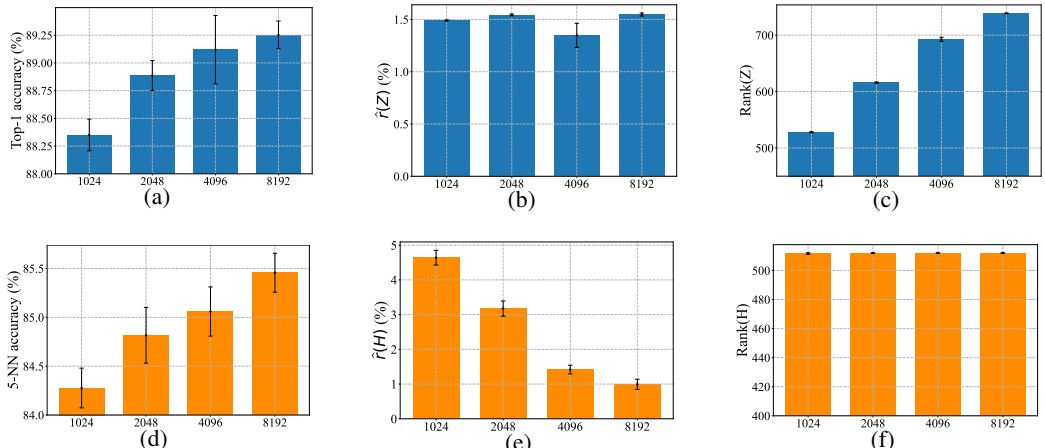

Figure II: Illustration of different projectors when varying the dimension of the hidden layer. We show (a) the linear accuracies; (b) the normalized stable-rank of *embedding*; (c) the rank of *embedding*. (d) the k-NN accuracies; (e) the normalized stable-rank of *encoding*; (f) the rank of *encoding*. All results are averaged by five random seeds, with standard deviation shown as error bars.

segmentation tasks. We use the default hyper-parameter configurations from the training scripts provided by the codebase for CW-RGP, using our 200-epoch pre-trained model on ImageNet.

For the experiments of '*VOC 07+12 detection*', we use Faster R-CNN fine-tuned in VOC 2007 trainval + 2012 train, evaluated in VOC 2007 test. For the experiments of '*COCO detection and COCO instance segmentation*', we use Mask R-CNN (1× schedule) fine-tuned in COCO 2017 train, evaluated in COCO 2017 val. All Faster/Mask R-CNN models are with the C4-backbone. Our CW-RGP is performed with 3 random seeds, with mean and standard deviation reported.

# E Investigating the Projector MLP

As mentioned in section 5 of the submitted paper, we conduct preliminary experiments to explore how the projector affects the extents of whitening between *encoding* and *embedding*. Specifically, we conduct experiments to explore the extents of whitening between *encoding* and *embedding* by varying the dimension and number of the hidden layer of the projector, based on our CW-RGP algorithm.

**Dimension of the Hidden Layer** Here, we conduct experiments on *CIFAR-10* to observe the effect by using different dimensions, ranging in $\{1024, 2048, 4096, 8192\}$, of the hidden layer. In the experiments, we set the projector: one hidden layer with BN and Relu applied to it and a linear layer as output (the *embedding* is 2048). We train the model for 200 epochs (other settings are the same as the experiments described in Section A.2). We use *encoding* as the representation for evaluation. The results are shown in Figure II. We observe that CW-RGP can obtain improved linear/5-NN accuracy, and increased rank of *embedding*, when increasing the dimension of the hidden layer. We find that CW-RGP can make the *encoding* full-rank for all settings with different dimensions. Besides, there are no significant differences in terms of the stable-rank (the extent of whitening) of *embedding* for all settings. One interesting observation is that the stable-rank of *encoding* decreases as the dimension of the hidden layer increases. For this observation, we conjecture that the large hidden-layer dimension may amplify the largest eigenvalue of the covariance matrix of *encoding* (driven by back-propagation), which leads to the decrease of stable-rank of *encoding*.

**Number of the Hidden Layer** We then conduct experiments on *Tiny ImageNet* to observe the effect by using different numbers, ranging in $\{1, 2, 3, 4, 5\}$, of the hidden layer (dimension of the hidden layer and embedding are 2048 and 1024 respectively). We train the model for 400 epochs (other settings are the same as the experiments described in Section A.2). We use *encoding* as the representation for evaluation. The results are shown in Figure III. We observe that CW-RGP can obtain increased rank and stable-rank (extent of whitening) of *embedding*, when decreasing the numbers of the hidden layer from 5 to 2. We also find that CW-RGP can make the *encoding* to be

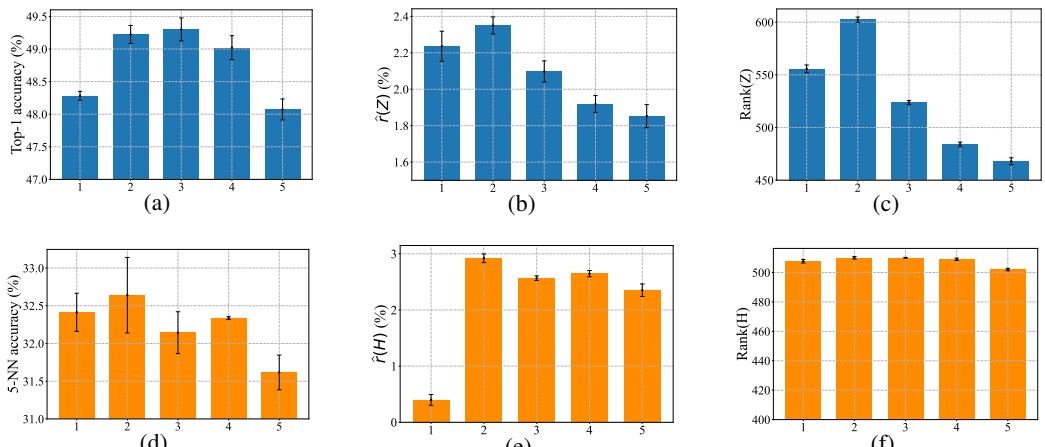

Figure III: Illustration of different projectors when varying the number of the hidden layer. We show (a) the linear accuracies; (b) the normalized stable-rank of *embedding*; (c) the rank of *embedding*. (d) the k-NN accuracies; (e) the normalized stable-rank of *encoding*; (f) the rank of *encoding*. All results are averaged by three random seeds, with standard deviation shown as error bars.

full-rank for almost all settings with different numbers of the hidden layer, except that CW-RGP with the projector using 5 hidden-layer has slightly reduced rank and has slightly worse performance. For this experiment, we do not obtain a clear clue that how the numbers of the hidden layer of the projector affects the extent of whitening.

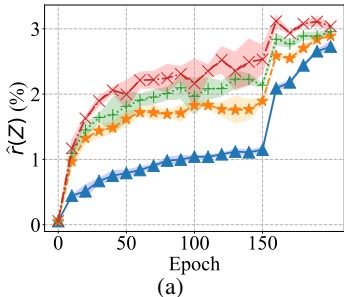 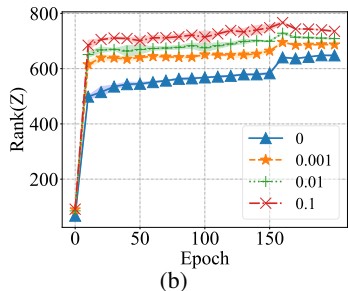

(a)          (b)

Figure IV: Illustration of CW with different penalty weights $\{0, 0.001, 0.01, 0.1\}$ of the covariance loss. We use the ResNet-18 as the encoder (dimension of representation is 512.), a two layer MLP with ReLU and BN appended as the projector (dimension of embedding is 2048). The model is trained on CIFAR-10 for 200 epochs with a batch size of 256 and standard data argumentation, using Adam optimizer [12] (more details of experimental setup please see Section A). We show (a) the normalized stable-rank of *embedding*; (b) the rank of *embedding*. The results are averaged by five random seeds, with standard deviation shown using shaded region.

# F    Covariance Loss along Channel Dimension

We note that VICReg [1] uses covariance loss along the batch dimension to constrain the evolution of the covariance matrix of *embedding* $\mathbf{Z}$ to a diagonal matrix. The main idea is to reduce the value of non-diagonal elements of the covariance matrix. It is natural to extend this covariance loss along the channel dimension and we explore whether covariance loss can be used together with whitening loss in this section.

For adapting to the sample orthogonalization like our channel whitening (CW), we propose a covariance loss along the channel dimension as follows:

$$Centering : \mathbf{Z}_c = (\mathbf{I} - \frac{1}{d}\mathbf{1} \cdot \mathbf{1}^T)\mathbf{Z}, \tag{6}$$

$$Covariance\ matrix : \Sigma = \frac{1}{d-1}\mathbf{Z}_c^T\mathbf{Z}_c, \tag{7}$$

$$Covariance\ loss : C = \frac{1}{m}\sum_{i \neq j}\Sigma_{i,j}^2, \tag{8}$$

We conduct experiments to show that our CW can work well with the covariance loss and the covariance loss even can amplify the extent of whitening of the embedding. We use CW combing the covariance loss with varying penalty weights ranging in $\{0, 0.001, 0.01, 0.1\}$. The results are shown in Figure IV. We observe that CW can obtain a significantly higher rank and stable-rank of *embedding* $\mathbf{Z}$, when increasing the penalty weight of covariance loss, especially in the early stage of training where a relatively large learning rate is used. We also note that CW with covariance loss using a penalty weight of 0.001 obtains an accuracy of $88.78\%$, slightly better than CW without covariance loss which has an accuracy of $88.50\%$.

# G    Licenses of Datasets

ImageNet [6] is subject to the ImageNet terms of access [5].

PASCAL VOC [8] uses images from Flickr, which is subject to the Flickr terms of use [9].

The annotations of COCO [15] are under the Creative Commons Attribution 4.0 License. The images are subject to the Flickr terms of use [9].