# OpenReview forum: "An Investigation into Whitening Loss for Self-supervised Learning"
_NeurIPS.cc/2022/Conference — NeurIPS 2022 Accept_

### Official Review · Reviewer_h596 · 2022-07-09

**Rating:** 7
**Confidence:** 3
**Soundness:** 4 excellent
**Presentation:** 4 excellent
**Contribution:** 4 excellent

**Summary:**

- Self-supervised learning approaches need to avoid collapse to a trivial representation. This has been tackled by various approaches in literature including use of negatives, use of asymmetric networks etc. Use of whitening loss has been explored by some recent works.
- This paper studies this whitening loss and various variants of the whitening transformation used in practice.
- The paper investigates some issues with the transformations used and a new random group partition based channel whitening approach to prevent collapse. The approach works well for large batch sizes which is shown through experiments on datasets like ImageNet and transfer to COCO.


**Questions:**

Please refer to the section on "Strengths And Weaknesses" for questions and comments.

**Limitations:**

- The authors have adequately discussed limitations of the proposed approach.
- While the work is more at a fundamental level, I do urge the authors to include some discussion on potential negative impacts.

**Strengths And Weaknesses:**

Strengths:
- The paper seeks to analyze various approaches used for whitening of feature representations used for SSL. The paper is very well written.
- The paper does a good job at explaining some of the preliminaries.
- Previous approaches have been analyzed extensively through experiments based on public repositories.
- The paper decomposes the whitening loss and connects it to common SSL approaches.
- Obtains state-of-the-art results on standard benchmark datasets.
- I think the paper will be of significance to the wider ML/SSL community.

Minor Weaknesses, suggestions and questions :
- Since one the Section 3.3 is one of the most important contributions of this paper, I would suggest having some more details & intuitions behind the explanations. Especially : L185-196. This could perhaps be included in the supplementary material.
- Can the centering+sharpening operation used in DINO & equipartion constraint used in SwAV be looked at through a similar analysis ?
- L280: To ensure $\frac{d}{g} > m$, do you modify the last layer of the projection head ?
- How does the proposed approach compare to state-of-the-art on training time ?
- Do you have some thoughts on why the gap in performance decreases at 100 ep vs 200 ep (Table 2) ? What about W-MSE 4 @ 200 ?
- I agree with the motivation in L263-266. I think from a practical standpoint, it is be a better idea to report required GPU memory & GPUs per approach in Table 2 in addition to the batch size.

---

> ### Author Response · Authors · 2022-08-02
> **Response to Reviewer h596**
>
> We thank the reviewer for the encouraging and insightful comments. Please find our responses to specific queries below.
>
> **Q1: Since one the Section 3.3 is ,..., supplementary material.**
>
> **A1:** Thanks for your suggestions. As recommended,  we will provide more details, e.g., in illustrating how minimizing Eqn.7 only requires the embedding being full-rank, and proving how the symmetric formulation (Eqn.5) can be equivalently decomposed into two asymmetric losses (Eqn.6) in the revised version.
>
> **Q2: Can the centering+sharpening ,..., a similar analysis ?**
>
> **A2:** We believe “the centering+sharpening operation” used in DINO, and “equipartition constraint” used in SwAV can be looked at through a similar analysis (connected to our whitening loss). The key point of our analysis is that the whitening loss in its symmetric formulation (Eqn. 5) can be decomposed into two asymmetric losses (Eqn. 6), where each asymmetric loss requires an online network to match a target with (whitened) constraints.
>
> 1. DINO explicitly formulates such a matching problem (online network to match target) from the view of knowledge distillation, and uses “centering+sharpening operation” to provides the constraints on the targets (e.g., it requires the targets to be centered, and requires that the scale of targets can be controlled by setting the temperature when using softmax). One significant difference between DINO and whitening loss is that DINO uses population statistics of centering (by moving average) while whitening loss uses the mini-batch statistics of whitening;
> 2. SwAV can also be regarded as two asymmetric losses (the swapped prediction used in SwAV), and each asymmetric loss can also be viewed as an online network (including the model and the learnable prototype matrix) to match a target with constraints (e.g, the equipartition constraint shown in Eqn. 4 of SwAV paper and the high-entropy constraints shown in Eqn. 3 of SwAV). Note that SwAV also explicitly uses “stop gradient” when it calculates the target (code) by using the iterative Sinkhorn-Knopp algorithm shown in the supplementary materials A1 of SwAV paper.
>
> Thanks for your insightful question.
>
> **Q3: L280: To ensure d/g >m, do you modify the last layer of the projection head  ?**
>
> **A3:** Yes, we modify the last layer of the projection head to ensure $d/g>m$. The dimension of output of the last layer of the projection head is $d$ (i.e., the dimension of the embedding described in the experimental setup of the paper). Note that the results in Table 1 of Shuffled-DBN has the same dimension ($d=512$) as our CW-RGP. We also have experiments on W-MSE with $d=512$ (using group necessary to ensure $m>d/g$), we find that it has worse performance (W-MSE-2 has 90.36% top-1 accuracy).
>
> **Q4: training time ?**
>
> **A4:** Our experiment on ImageNet using the ResNet-50 is run on a workstation with only 4 GPUs (A6000 with 48G memory), and it takes roughly 96 hours (4 days) in training for 100 epochs. Even though whitening transformation has the time complexity of $O(dm^2+m^3)$, it has much less computation cost, compared to the computation cost of the backbone (encoder). E.g., on the experimental setup (ResNet-18, batch size of 256) for CIFAR-10, CW-RGP costs 23.09 s/epoch, while BYOL and Shuffled-DBN costs 25.21s/epoch and 24.78s/epoch, respectively.
>
> **Q5: Do you have some thoughts on why the gap in performance decreases at 100 ep vs 200 ep (Table 2) ? What about W-MSE 4 @ 200 ?**
>
> **A5:**  In terms of the decrease of gap in performance at 100 epoch vs 200 epoch, we conjecture the reason is likely that CW-RGP has advantage in improving the efficiency of the learning for self-supervised learning, because it provides whitened targets, which provides non-redundant signal in learning representation. We observe this phenomena base on our preliminary experiment, by monitoring the training progress. This advantage is much more important in the early training.
>
> The result of W-MSE 4@100 in Table 2 is reported from the W-MSE paper [12]. We donot reproduce W-MSE on ImageNet, because it requires a large batch size (bs=2048) and our workstation with 4GPUs does not have enough memory to run the experiments with such large a batch size. The experiments for reproducing W-MSE are currently in the pipeline and a through comparison will be included in the revised version.
>
> **Q6: Should report GPU memory in Table 2.**
>
> **A6:** Thanks sincerely for your constructive suggestions! We use a workstation with 4 NVIDIA RTX A6000(48G) for our method on ImageNet. Our method CW-RGP 4 required about 4*45G GPU memory when training. The other results shown in Table 2 are mostly reported in the SimSiam paper[8], except that the result of W-MSE 4 is from the W-MSE paper [12], while GPU memory & GPUs are not reported in these papers.

---

> > ### Comment · Reviewer_h596 · 2022-08-07
> > **Thanks for the rebuttal**
> >
> > Thanks for the detailed rebuttal. It adequately answers my questions. I think it will be good to include the relevant discussions in the next draft.
> > I do not have any new concerns with this paper.

---

### Official Review · Reviewer_qKzP · 2022-07-10

**Rating:** 7
**Confidence:** 4
**Soundness:** 3 good
**Presentation:** 3 good
**Contribution:** 3 good

**Summary:**

This paper gives out a thorough investigation into whitening loss applied in self-supervised learning. Based on their analysis, they proposed a channel whitening method named CW-RGP. From their experiments on variuos datasets, CW-RGP gets the SOTA in most cases.

**Questions:**

I would wonder whether the Barlow Twins method can be better.

Then grouping the experiments based on their techniques is easier to show the contribution, especially this paper belongs to the whitening loss branch.

Finally, the comparison methods from Table 1,2,3 are a bit different, actually I was finding the W-MSE results in Table 3. Can you explain a bit?

**Ethics Review Area:**

["I don’t know"]

**Strengths And Weaknesses:**

Strengths:

1, The writing is nicely done and the paper is organized well for understanding.

2, The experiments are quite sufficient, including the analysis experiments and comparison experiments.

3, The understanding for whitening loss is convincing and there are also some new indicators proposed for future study.

Weakness:

Overall, I didn't see much disadvantages in this paper, I'm just curious about several points below.

1, The barlow twins and VICReg methods are not present in all tables. In the related work, the authors have classified them as soft whitening, so I think some comparison with them should not be neglected.

2, Although, Barlow Twins might get even better results than CW-RGP (from my experiments before), I don't think it can be very important. This paper mainly studies the whitening loss, thus I think the comparison with whitening loss based method (such as W-MSE) is more important.

---

> ### Author Response · Authors · 2022-08-02
> **Response to Reviewer qKzP**
>
> We thank the reviewer for the encouraging and insightful comments. Please find our responses to specific questions below.
>
> **Question 1: I would wonder whether the Barlow Twins method can be better. Then grouping the experiments based on their techniques is easier to show the contribution, especially this paper belongs to the whitening loss branch.**
>
>
>
> **Response:** Thanks sincerely for your constructive suggestions.
> We note that the Barlow Twins paper does not provide the SSL trained results of Barlow Twins on CIFAR/STL-10/TinyImageNet. We thus conduct additional experiments for Barlow Twins, following the same setup in Table 1, and we use the default hyper-parameters and evaluation protocol as in W-MSE [12] (e.g., trained with Adam optimizer). We use lambda=0.0051 (recommend in Barlow Twins), provide three setups with different projector ({1024-1024, 2048-2048 and 2048-2048-2048}), we report the best results from the three setups. Barlow Twins has top-1 accuracy {87.51, 60.71, 84.55 and 41.49} and 5-nn accuracy {84.19, 54.95, 80.21, 31.84} on CIFAR-10, CIFAR-100, STL-10 and Tiny-ImageNet, respectively. We notice that by further finetuning the lambda hyper-parameter on CIFAR-10, the results of Barlow twins improve to 88.51% top-1 accuracy and 86.53% 5-nn accuracy. It is likely that we donot well reproduce Barlow Twins, since the recommended hyper-parameters of Barlow Twins is for the training of ImageNet. We would like to include them in the revised manuscript. As also mentioned by the reviewer, our manuscript mainly studies the whitening loss and presents the comparison with whitening loss based method (such as W-MSE).
>
>
>
> **Question 2: Finally, the comparison methods from Table 1,2,3 are a bit different, actually I was finding the W-MSE results in Table 3. Can you explain a bit?**
>
> **Response:** The experiments of Table 1 (CIFAR-10/100, STL-10 and TinyImageNet) are based on the public released code of W-MSE paper [12]. We reproduce the results of W-MSE and other baselines, which are also used in the W-MSE paper (except that we reproduce the following SimSiam and Shuffled-DBN method). However, the released code of W-MSE paper [12] does not provide the experimental setups for ImageNet. We thus use the released code of SimSiam paper [8], which is regarded as a good codebase for the experiments on ImageNet (Table 2). We further transfer the model trained on ImageNet to the object detection and instance segmentation tasks (Table 3). Therefore, the results of baselines on ImageNet (Table 2) and Transfer Learning (Table 3) are inherited from the SimSiam paper [8].
> We note that we report the result of W-MSE@100 epoch, which is from the W-MSE paper. We donot reproduce W-MSE on ImageNet, because it requires a large batch size (bs=2048) and our workstation with 4GPUs does not have enough memory to run the experiments with such large a batch size.  Therefore, we could also not perform downstream tasks without the pretraining W-MSE model on ImageNet (W-MSE paper does not provide the pre-trained W-MSE model and also not show the results transferring to object detection tasks). The experiments for reproducing W-MSE are currently in the pipeline and a through comparison will be included in the revised version.

---

> > ### Comment · Reviewer_qKzP · 2022-08-09
> > **Thanks for the rebuttal**
> >
> > Thanks for the rebuttal, actually I don't have a lot of hesitation to accept this paper with well writing and sufficient experiments. If only 4 GPUs is the truth, I hope in the future with released code, there can be future job to make it finished. I still hold my rating as 7. Accept.

---

### Official Review · Reviewer_j1ii · 2022-07-10

**Rating:** 5
**Confidence:** 4
**Soundness:** 2 fair
**Presentation:** 2 fair
**Contribution:** 3 good

**Summary:**

The paper provides an analysis of different whitening losses used in Self-supervised learning, seeking to interpret some empirical observations, e.g. the connection between whitening losses and the asymmetric methods (BYOL/SimSiam), why PCA does not work, and why whitened outputs are not good representations. The paper also proposes channel whitening with random group partition (CW-RGP), which is shown to be an effective whitening loss. CW-RGP is evaluated on image recognition (ImageNet1k and 4 other benchmarks), object detection (VOC/COCO), and instance segmentation (COCO).

**Questions:**

i) When evaluating the performance of the whitened representation (L167, Fig.3) I wonder if the whitening matrix (\Phi(Z)) is estimated per mini-batch or over the whole training set?

ii) L308-309, are “d=2” and “d=4” here referring to “g=2” and “g=4”?


**Limitations:**

Limitations are discussed in the main paper. Potential negative societal impacts are not discussed.

**Strengths And Weaknesses:**

Strengths

i) the proposed random group partition is technically sound;

ii) the performance of the proposed CW-RGP method looks promising on some benchmarks e.g. COCO object detection/ instance segmentation;

iii) an ablation study on the batch size is provided;

iv) the writing is clear in general.


Weaknesses

i)  The interpretations about “why PCA does not work” and “why the whitened output is not good” are not convincing. To me, the explanation could be much simpler and more intuitive: the batch whitened outputs rely on the batch statistics: an image may have different whitened representations when computed in different mini-batches.

When using PCA, the descriptor of an image relies on the eigenvectors (U in L136), which may change dramatically across mini-batches. This explains why BW-based approaches prefer large batch sizes. It explains the experimental results shown in Fig. 4. It also explains why whitened outputs are not good representations, i.e. experiments in Fig. 3.

Note that the predictors in the asymmetric methods (L197), on the other hand, do not rely on batch statistics, which I believe is a key difference.

ii) The comparisons in Table 2 may not show the full picture, e.g. baselines like BYOL/SWAV may be significantly under-trained. Here, the batch size for BYOL/SWAV is 4096. When trained for fewer epochs (e.g. 200 epochs), a large batch size may hurt the performance as it leads to significantly fewer training iterations. It would be better if baselines like BYOL/SWAV-batch-size-512-epoch-100/200 are also included.


iv) Channel whitening has been proposed before in [47] for the same task. As [47] has been published in ICLR 2022. I’m not sure if [47] could be considered a concurrent work. Compared to [47], the new content is the random group partition. This extra design may not be enough for NeurIPS. Overall, I believe [47] should at least be included as a baseline, and the ablation on the random group partition should be included.

Minor issues

i) L18-19 “two networks are trained …[8]”, I think there is only one network

ii) L286: “rand” → “random”

iii) Table 1, Simsim → SimSiam

iv) Table 1, references are included for some baselines (SimCLR, BYOL), but not all (e.g. Shuffled-DBN, W-MSE)

v) Table 2 & 3, it would be better if references are included.

---

> ### Author Response · Authors · 2022-08-02
> **Response to Reviewer j1ii (1/2)**
>
> We thank the reviewer for the encouraging and insightful comments. Please find our responses to specific queries (questions and concerns) below.
>
> **Question 1: When evaluating the performance of the whitened representation (L167, Fig.3) I wonder if the whitening matrix $\phi(Z)$ is estimated per mini-batch or over the whole training set?**
>
> **Response:**  In Fig.3, the whitening matrix is calculated over the whole training set, when we evaluate the performance of the whitened representation. In this way, we ensure that the output is whitened with normalized stable rank=100%, as pointed out in Fig. 3.
> We also perform experiments using the estimated whitening matrix using running averages over different min-batch statistics (the so-called evaluation mode in BN/BW). In this way, we also observe that the whitened output has worse performance than the encoding and embedding, with a significant margin. Note that using the estimated whitening matrix can also obtain a high normalized stable-rank (e.g, 99%) after the training is finished.
>
> **Question 2: L308-309, are “d=2” and “d=4” here referring to “g=2” and “g=4”?**
>
> **Response:** “d=2” and “d=4” is not referred to “g=2” and “g=4” here.  “d=2” and “d=4” indicate that 2 and 4 positives views are extracted per image respectively, similar to W-MSE [12]. Considering that this may be confused with the size of channels $d$ in L 263-295, we will replace $d$ with $s$ in the revised version. Thank you for pointing it out.
>
> **Concern 1: Regarding The interpretations about “why PCA does not work” and “why the whitened output is not good”.**
>
> **Response:** We first thank the reviewer for the detailed comment. We hope that reviewer recognizes our analysis that the whitening loss in its symmetric formulation (Eqn. 5) can be decomposed into two asymmetric losses (Eqn. 6), where each asymmetric loss requires an online network to match a target with (whitened) constraints. From Eqn. 6, we can clearly find that the target $\hat{Z}$ and the whitening transformation $\phi(Z)$ are varying for each mini-batch. Therefore, Eqn. 5 implies the explanation from the reviewer that “an image may have different whitened representations when computed in different mini-batches.” And “the descriptor of an image relies on the eigenvectors ($U$ in L136), which may change dramatically across mini-batches”. However, we would like to point out that Eqn.6 (and the explanation suggested by the reviewer) is not sufficient to explain “why PCA does not work”. Since ZCA whitening has both varying target $\hat{Z}$ and varying whitening transformation $\phi(Z)$ (ZCA whitening also depends on the $U$) over different mini-batches, the question is why ZCA whitening can work? (if the suggested explanation is true). Apparently, we need to further compare the magnitude of diversity among different mini-batches. That is why we show Fig. 4, which shows that PCA whitening has significantly diverse targets and whitening transformations. Therefore, it is difficult for the online network to match such a target signal with significant variation, resulting in minimal decrease in the whitening loss, which explains why PCA whitening does not work well. We thank the reviewer and will further improve the text in the revised manuscript.
>
>
> **Concern 2: Regarding the comparisons in Table 2 (e.g. baselines like BYOL/SWAV).**
>
> **Response:** We thank the reviewer for the suggestion. Please note that the results of BYOL/SwAV in Table 2 is from the SimSiam paper [8], since our experiments are based on the codebase of SimSiam paper [8] and our workstation with 4 GPUs (RTX A6000) can not reproduce BYOL/SwAV due to the limitation of memory. We note that BYOL has degenerated performance when decreasing the batch size, as shown in Fig.3 of the BYOL paper [16] (e.g., BYOL with batch size of 256 has near 2.6% drop in top-1 accuracy, compare to the one with batch size of 4096). As recommended, we will further experiment by running BYOL/SWAV with the batch size of 512 and include the results in the revised manuscript.
>
>
> **Please find the response of the remaining comments (Concern 3 and "Minor issues") in the next comment box (2/2).**

---

> > ### Author Response · Authors · 2022-08-02
> > **Response to Reviewer j1ii (2/2)**
> >
> >
> > **Concern 3: Comparison with [47] (ICLR 2022) and additional ablation on the random group partition.**
> >
> > **Response:** We would like to highlight that the proposed instance-wise contrastive learning (ICL) in [47] is not the same as our mentioned CW only method (without RGP). There are several significant differences in technical details which we note by delving into the details of implementation (from the official codebase of [47]) and the rebuttal/discussion regarding [47] on OpenReview.
> >
> > 1. ICL in [47] uses “stop-gradient” for the whitening matrix (whitening matrix is viewed as a set of parameters rather than a function over mini-batch data), which can be viewed as a naïve linear transformation, while our whitening matrix is viewed as a function that requires back-propagation through the whitening transformation. Similarly, in this way, Feature-wise Contrastive learning (FCL) in [47] is not BW used in W-MSE/Shuffled-DBN.
> >
> > 2. ICL in [47] uses extra pre-conditioning $\alpha I$ on the covariance matrix, i.e., $\Sigma^{'} = (1-\alpha)*\Sigma + \alpha I$, where $\alpha = 0.1$. Furthermore, ICL also uses extra pre-conditioning $λI$ on the whitening matrix, i.e.,  $\phi(Z) = U(\Lambda + λI)^{-1/2}$,  where $λ = 10^{-5}$. CW does not use extra pre-conditioning on the covariance matrix and whitening matrix. We experimentally observe that ICL suffers numerical instability when we remove the pre-conditioning (setting  as zero or very small number), while our CW can work well without pre-conditioning since that the loss (Eqn.5) with back-propagation through the whitening transformation used in CW encourages the embedding $Z$ to be full-rank.
> >
> > 3. ICL in [47] uses the invariance loss of Barlow Twins [44] which can prevent full collapse by encouraging the diagonal elements of covariance matrix to be $1$ (like BN), while CW/BW(W-MSE, Shuffled-DBN) use the common MSE or normalized MSE which only minimizes the distances between different views without extra constraints. We experimentally observe that ICL suffers severe (dimensional) collapse if we use MSE (or normalized MSE).
> >
> >    We will clarify the differences of CW and ICL in [47] in the revised version.
> >
> >   In terms of the comparison in performance, our CW-RGP can obtain better performance than Zero-CL in [47] (that combine ICL and FCL), e.g., on CIFAR-10, Zero-CL has 90.81% top-1 accuracy while our CW-RGP has 92.47% top-1 accuracy, when training for 1000 epochs using ResNet-18; on ImageNet, Zero-CL has 68.9% top-1 accuracy while our CW-RGP has 69.7% top-1 accuracy, when training for 100 epochs using ResNet-50. We would add the results of [47] in the revised version.
> >
> > In terms of the ablation study for Random Group Partition (RGP), we conducted the preliminary comparison in Figure 6 of the paper. Here, we conducted additional experiments on CIFAR-10 & CIFAR-100 to further show the effectiveness of RGP, following the setting in Table 1. The results are reported in the following table, which clearly shows the advantages of RGP.
> >
> >
> >
> > |    Method    |    &nbsp;&nbsp;&nbsp;&nbsp; CIFAR-10     |        &nbsp;&nbsp;&nbsp;&nbsp; &nbsp;CIFAR-100        |
> > | :----------: | :--------------------------------------: | :----------------------------------------------------: |
> > |              |      top-1 &nbsp;&nbsp;&nbsp; 5-nn       |       &nbsp; &nbsp;top-1 &nbsp;&nbsp;&nbsp; 5-nn       |
> > |     CW 2     |      91.66 &nbsp;&nbsp;&nbsp; 88.99      |     &nbsp; &nbsp;66.26  &nbsp;&nbsp;&nbsp;  56.36      |
> > |   CW-GP 2    |     91.61  &nbsp;&nbsp;&nbsp;  88.89     |     &nbsp; &nbsp; 66.17  &nbsp;&nbsp;&nbsp;  56.53     |
> > | **CW-RGP 2** | **91.92** &nbsp;&nbsp;&nbsp;   **89.54** | &nbsp; &nbsp; **67.51** &nbsp;&nbsp;&nbsp;   **57.35** |
> > |              |                                          |                                                        |
> > |     CW 4     |     92.10 &nbsp;&nbsp;&nbsp;  90.12      |     &nbsp; &nbsp; 66.90 &nbsp;&nbsp;&nbsp;  57.12      |
> > |   CW-GP 4    |     92.08  &nbsp;&nbsp;&nbsp; 90.06      |     &nbsp; &nbsp; 67.34 &nbsp;&nbsp;&nbsp;  57.28      |
> > | **CW-RGP 4** |  **92.47** &nbsp;&nbsp;&nbsp; **90.74**  | &nbsp; &nbsp; **68.26** &nbsp;&nbsp;&nbsp;  **58.67**  |
> >
> >
> >
> > **Minor issues:** We thank the reviewer and will fix all these in the revised manuscript.

---

> > > ### Comment · Reviewer_j1ii · 2022-08-09
> > > **Comment after the rebuttal**
> > >
> > > Thank you for the rebuttal. It resolves some of my concerns (question 1,2, concern 3). I’d like to raise the score to 5. To me, oscillating signal across mini-batches seems to be a simpler/better explanation for the failures of some of the whitening losses (confirmed by the PCA experiments presented in the paper). The comparisons in Table 2 are also not fully convincing as the settings of the models are not the same: with a batch-size of 4096, BYOL and SWAV were trained with 8x fewer iterations than the proposed method. They can be further explored in future research.

---

### Author Response · Authors · 2022-08-02
**Thank you for the valuable comments**

We thank all the reviewers (j1ii, qKzP, h596) for their detailed and positive feedback: writing is clear and nicely done (j1ii, qKzP, h596), the proposed random group partition is technically sound (j1ii), understanding for whitening loss is convincing (qKzP), experiments are quite sufficient (qKzP), the paper will be of significance to the wider ML/SSL community (h596), obtains state-of-the-art results on standard benchmark datasets (h596).

Our codes and trained models will be publicly released.

---

### Meta-Review · Area_Chair_XsRq · 2022-08-24

**Recommendation:** Accept
**Confidence:** Certain

**Metareview:**

This paper studies the impact of whitening losses used in recent Self-supervised learning (SSL) methods. It shows that the symmetric whitening loss can be decomposed into two asymmetric losses, explaining important behaviour experimentally observed (*e.g.* why some whitening transformations -*e.g.* PCA- are not always effective, and why whitened output is not always a good representation). The authors proposed a channel whitening with random group partition (CW-RGP) with good experimental performances.

The paper initially received mixed reviews: 2 borderline reject and 2 accept recommendations. The main reviewers concerns related to the soundness of the proposed analysis of the whitening loss, the fairness of the experiments, and the positioning and comparison to other recent baselines (*e.g.* [47], DINO or SwAV). The rebuttal did a good job in answering the reviewers' comments, and there was a consensus that the paper should be accepted after the rebuttal.

The AC's owns reading of the submission confirmed that the paper is a solid contribution for NeurIPS. It considers that the proposed analysis of whitening loss is valuable for the community, and that the proposed CW-RGP is meaningful and experimentally validated. Thus, the AC recommends acceptance.
The reviewer's consensus for acceptance has been reached given the authors' feedback including promises to improve the clarity and insights of the analysis, and regarding the positioning with recent baselines and new experimental results. Therefore, the authors are highly encouraged to carefully include these improvements into the final paper.


**Award:**

No

---

### Decision · Program_Chairs · 2022-09-14

Accept